# Seeding the meiotic DNA break machinery and initiating recombination on chromosome axes

Ihsan Dereli [1], Vladyslav Telychko[1], Frantzeskos Papanikos[1], Kavya Raveendran[1], Jiaqi Xu[2,3], Michiel Boekhout [2], Marcello Stanzione[1], Benjamin Neuditschko[4], Naga Sailaja Imjeti[1], Elizaveta Selezneva[5], Hasibe Tuncay [6], Sevgican Demir[1], Teresa Giannattasio [7], Marc Gentzel [8], Anastasiia Bondarieva[1], Michelle Stevense[1], Marco Barchi [7,9], Arp Schnittger [6], John R. Weir [5], Franz Herzog [4], Scott Keeney[2,3,10] & Attila Tóth [1] ✉

Programmed DNA double-strand break (DSB) formation is a crucial feature of meiosis in most organisms. DSBs initiate recombination-mediated linking of homologous chromosomes, which enables correct chromosome segregation in meiosis. DSBs are generated on chromosome axes by heterooligomeric focal clusters of DSB-factors. Whereas DNA-driven protein condensation is thought to assemble the DSB-machinery, its targeting to chromosome axes is poorly understood. We uncover in mice that efficient biogenesis of DSB-machinery clusters requires seeding by axial IHO1 platforms. Both IHO1 phosphorylation and formation of axial IHO1 platforms are diminished by chemical inhibition of DBF4-dependent kinase (DDK), suggesting that DDK contributes to the control of the axial DSB-machinery. Furthermore, we show that axial IHO1 platforms are based on an interaction between IHO1 and the chromosomal axis component HORMAD1. IHO1-HORMAD1-mediated seeding of the DSB-machinery on axes ensures sufficiency of DSBs for efficient pairing of homologous chromosomes. Without IHO1-HORMAD1 interaction, residual DSBs depend on ANKRD31, which enhances both the seeding and the growth of DSB-machinery clusters. Thus, recombination initiation is ensured by complementary pathways that differentially support seeding and growth of DSB-machinery clusters, thereby synergistically enabling DSB-machinery condensation on chromosomal axes.

Sexually reproducing eukaryotes employ meiosis to generate haploid reproductive cells from diploid mother cells. One of the key features of meiosis is a specialized homologous recombination that is initiated by programmed formation of DNA double strand breaks (DSBs) at the onset of the first meiotic prophase (reviewed in[1]). Meiotic DSBs are generated by a type II topoisomerase-related enzyme complex consisting of a catalytic subunit, SPO11, and a co-factor, TOPOVIBL[2–7].

Meiotic recombination leads to the juxtaposition/synapsis of homologous copies (homologs) of each chromosome in synaptonemal complexes (SCs). Recombination repairs DSBs within the context of synapsed chromosomes, thereby generating inter-homolog genetic exchanges, which produce new allele combinations. Furthermore, a subset of the exchanges mature into chromosomal crossovers (COs), which form the basis of inter-homolog linkages that enable orderly

halving of chromosome numbers in meiosis. Given the potential genotoxicity of DSBs and the importance of COs, both DSB formation and repair are under tight spatiotemporal control[8].

Chromosomes are organized into linear arrays of DNA loops that are anchored on proteinaceous chromosomal cores, called axes, in meiosis[9]. DSBs are spatially restricted to chromosome axes, which is thought to stem from the concentration of SPO11-activating proteins on axes. SPO11 activity requires several auxiliary proteins in most eukaryotes (reviewed in[10,11], summarized in Supplementary Fig. 1). While there is considerable divergence in DSB factors, three SPO11 auxiliary protein families − represented by the budding yeast (*Sc*) Mer2, Mei4 and Rec114 − are conserved in diverse clades of fungi, plants and animals[12–16]. Both the budding yeast proteins[17–20] and the corresponding mouse (*Mm*) proteins (IHO1 (Mer2 ortholog[14]), MEI4 and REC114) jointly form axis-bound focal clusters (hereafter DSB-factor clusters) that are hypothesized to enable SPO11 activity[13,21,22]. Mammalian DSB-factor clusters incorporate at least two further components, ANKRD31[23,24], which seems specific to vertebrates, and MEI1[25], orthologs of which are currently known in vertebrates and plants[26,27]. In vitro studies suggest that DSB factors form chromatin-bound clusters by DNA-driven protein condensation which relies on multivalent protein-protein and protein-DNA interactions[28]. However, the mechanisms targeting DSB-factor clustering to chromosome axes in vivo are not clear.

Axial accumulation of the DSB-machinery was proposed to partially depend on interactions between Mer2/IHO1-family proteins and conserved axis-associated HORMA domain proteins in diverse taxa including fungi[20,29,30], plants[31] and mammals[22]. In mammals, the HORMA domain protein HORMAD1 is hypothesized to enhance DSB activity[32,33] by enabling the formation of extended axial IHO1 platforms, which serve as substructures for focal clusters of SPO11 auxiliary proteins[22]. Consistent with this hypothesis, (1) axial IHO1 depends on HORMAD1, but not on SPO11 auxiliary proteins[21,22], (2) axial IHO1 accumulations extend beyond the boundaries of focal DSB-factor clusters[22], (3) focal DSB-factor clusters largely depend on IHO1 and, to a lower extent, HORMAD1 in most of the genome[22,24], and (4) DSBs depend fully on IHO1[22] and partly on HORMAD1[32,33]. In line with these observations, HORMAD1 regulation is thought to enable correct spatiotemporal patterning of DSB activity. In several studied models, SC seems to limit DSB-machinery[13,18,20,22,31,34–36] and DSB activity[8,37–39] to unsynapsed sections of axes where DSBs are used for homolog pairing and synapsis. This regulatory mechanism was hypothesized to involve an SC-triggered depletion of HORMAD1 from synapsed axes in mammals[40].

DSBs have alternative requirements for SPO11 auxiliary proteins in the relatively short (~0.7 Mb in mice) pseudoautosomal regions (PARs) of sex chromosomes, where DSB-factor levels[23–25] and DSB activity[41] are greatly elevated to enable X and Y chromosome pairing in males. Whereas ANKRD31 is not essential for DSB−factor clusters and DSBs in most of the genome, ANKRD31 is critical in the PAR[23,24]. Neither IHO1[24] nor HORMAD1[25] is needed for enrichment of SPO11 auxiliary proteins on PAR axes.

Previous work has revealed the importance of HORMAD1 and IHO1 in the formation of DSB-machinery clusters (see previous paragraphs), however it remains unknown if and how the HORMAD1-IHO1 interaction enables assembly of DSB-machinery on axes. Here, we reveal that the C-terminal 7 amino acids of IHO1 are required for (1) IHO1-HORMAD1 interaction, (2) the formation of axial IHO1 platforms and (3) efficient seeding of cytologically distinguishable DSB-factor clusters. These observations collectively suggest that seeding of the DSB machinery on axes critically complements and enhances the previously suggested mechanism of DSB-factor clustering by DNA-driven condensation in vivo. We also discover that whereas IHO1-HORMAD1 interaction specifically enhances seeding of DSB-factor clusters, ANKRD31 supports both seeding and growth. The IHO1-

HORMAD1 complex and ANKRD31 act synergistically − their simultaneous disruption abolishes DSB-factor clusters and DSBs. Thus, DSB formation is enabled on chromosome axes by complementary pathways that differ in both mechanism and preferred genomic locations.

## Results

### IHO1-HORMAD1 interaction requires a conserved acidic motif in the IHO1 C-terminus

A direct HORMAD1-IHO1 interaction may enable focusing of DSB activity to unsynapsed axes[22]. Therefore, we mapped HORMAD1-interacting regions of IHO1 in yeast two-hybrid (Y2H) assays (Fig. 1A, B, Supplementary Table 1). The first 358 (of 574) amino acids of IHO1, including a conserved coil domain, were neither sufficient nor required for interaction with HORMAD1. In contrast, IHO1 fragments that included the C-terminal 75 amino acids of IHO1 interacted with HORMAD1 and, specifically, its HORMA domain. Further, in vitro binding assays reconfirmed efficient interaction between the HORMA domain of HORMAD1 and the C-terminal 215 amino acids of IHO1 (Supplementary Fig. 2A, B).

IHO1 is phosphorylated in vivo on at least four positions, S476, T490, S569 and S570 ([42], see https://phosphomouse.hms.harvard.edu/site_view.php?ref=IPI00914141), which are also located in the C-terminal region of IHO1 (Fig. 1A). Simultaneous replacement of S476 and T490 of IHO1 by alanine did not affect the Y2H interaction of IHO1 and HORMAD1. In contrast, IHO1-HORMAD1 interactions were abolished by (i) simultaneous exchange of serines to alanines in positions 569 and 570 (Fig. 1A, B) or (ii) the deletion of the last 7 amino acids of IHO1 (hereafter IHO1_C7Δ, Fig. 1A, B, Supplementary Fig. 2A, B, Supplementary Table 1). All known IHO1 interactors except HORMAD1 efficiently interacted with IHO1_C7Δ (Fig. 1B, Supplementary Table 1), supporting the idea that the C-terminus of IHO1 is specifically important for interaction with HORMAD1 (via the HORMA domain).

S569 and S570 of IHO1 are located in an acidic $FDS_{(569)}S_{(570)}DDD$ sequence patch that overlaps with the C-terminal end of a widely conserved short similarity motif (called SSM2) of Mer2/IHO1-family proteins[14] (Fig. 1A, Supplementary Fig. 2C). Clusters of acidic residues are almost universally present in or next to SSM2s, and one or more serine/threonine(s) are often found in the acidic patch, particularly in vertebrates and plants (ref. [14], Supplementary Fig. 2C and Supplementary Table 2).

Consistent with their conservation, the C-termini of Mer2/IHO1-family proteins are important for interactions with HORMAD1 orthologs in several taxa, including budding[29] and fission[30] yeasts in addition to mammals (Supplementary Table 2). IHO1 and HORMAD1 orthologs also interact in plants[31,36,43,44]. In *Arabidopsis thaliana* (*At*), the coiled coil-containing N-terminus of *At*Mer2/IHO1 (PRD3) − but not the SSM2-harboring C-terminus − was reported to interact with the SWIRM domain-harboring C-terminus of the *At*HORMAD1 (ASY1)[36]. In contrast, we found that the C-terminus of *At*PRD3 interacted with the HORMA domain-containing N-terminus of *At*ASY1 in low stringency Y2H, and that the interaction required the SSM2-linked acidic patch of *At*PRD3 (Supplementary Fig. 2D). Thus, conserved acidic patches associated with SSM2s may enable and/or enhance interaction of IHO1- and HORMAD1-related proteins in diverse taxa, albeit the importance and the molecular role of SSM2/acidic patches may differ between species (summarized in Supplementary Table 2).

### IHO1-axis association requires the IHO1 C-terminus

To test if the C-terminal region of IHO1 was important for IHO1 localization to chromosomes we ectopically expressed GFP fusions of wild type and mutant versions of IHO1 in spermatocytes by in vivo electroporation of mouse testes (Fig. 1C, D). IHO1 mutations included a deletion of the last 7 amino acids, or an exchange of S569 and/or S570 for either non-phosphorylatable alanine, or phosphomimetic aspartates or glutamates. All of the tested mutations impaired axial

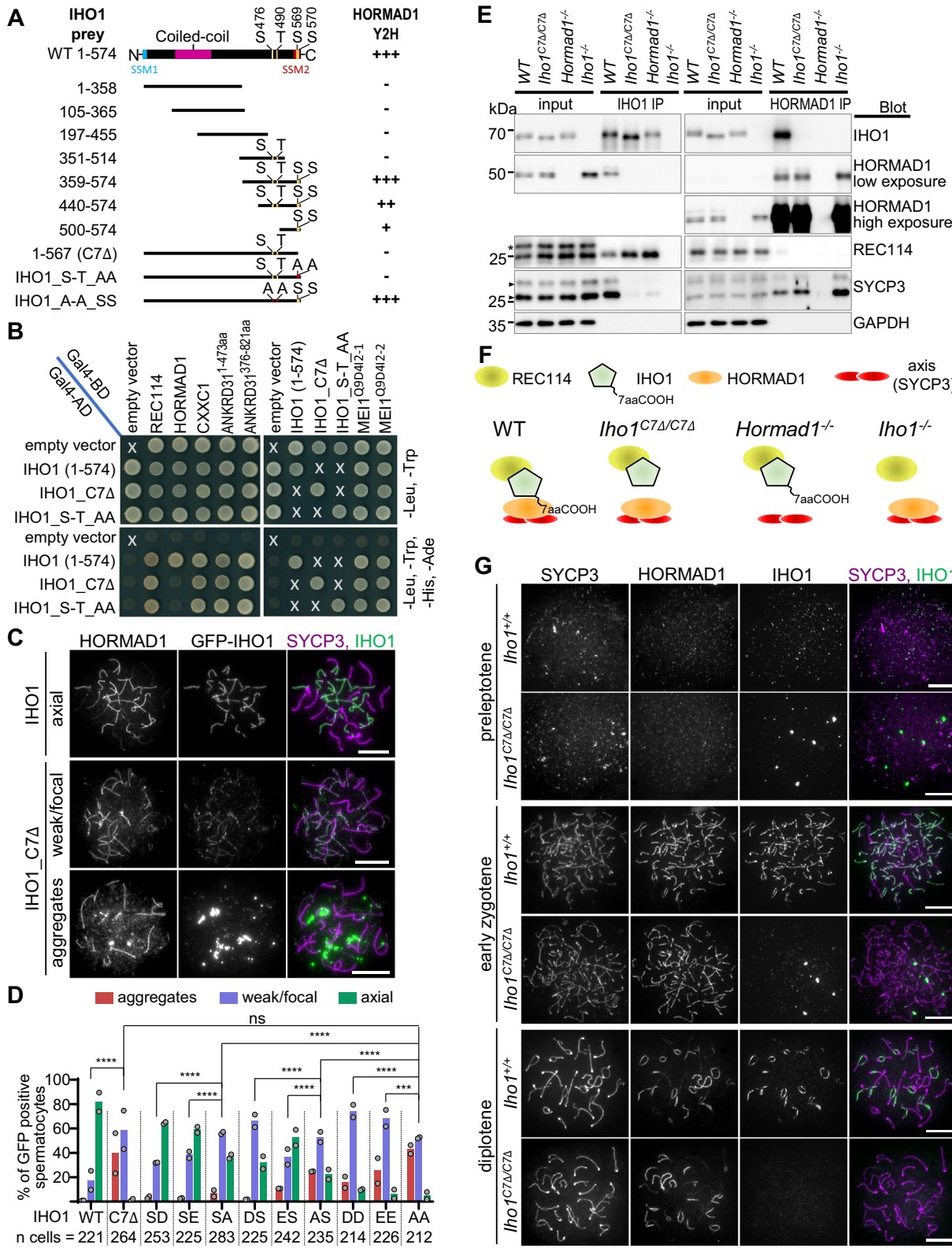

localization of the GFP-IHO1 fusions, but serine to alanine mutations resulted in more severe defects than serine to aspartate or glutamate exchanges in respective positions. IHO1 localization was most severely disrupted if either the last 7 amino acids were deleted or both S569 and S570 were exchanged with alanines. These versions of IHO1 rarely accumulated effectively on axes, although weak focal staining on chromatin and/or aggregate formation was often observed. IHO1

C-terminal mutations caused broadly matching disruptions in axial IHO1 accumulation in vivo (Fig. 1C, D) and IHO1-HORMAD1 interactions in Y2H assays (Supplementary Table 1). Together, these observations suggest that the IHO1 C-terminus (in particular S569 and S570) enhances axial recruitment of IHO1 by promoting IHO1-HORMAD1 interaction. According to phenotypes caused by phosphomimetic replacements of S569 and S570, phosphorylation at these sites may

**Fig. 1 | IHO1 C-terminus mediates IHO1-HORMAD1 interaction. A, B** Y2H assays between IHO1 interactors (this study and[22,24,113]) and wild-type (1-574) or modified versions of IHO1. **A** Schematics show conserved domains[14] and positions of phospho-serines (S) or -threonine (T) or their substitution with alanine (A) in IHO1. **B** Budding yeast cultures co-transformed with indicated pairs of Y2H baits (top) and preys (left side) are shown after 3 (two left images) or 2 (two right images) days of growth on drop-out plates. X marks bait-prey combinations that were omitted from Y2H due to lack of relevance. **C, G** Immunostaining in nuclear spread spermatocytes of 13 days postpartum (dpp) (**C**) and adult (**G**) mice. Chromosome axis (SYCP3, **C** overlay, **G**), HORMAD1 (**C, G**) and either ectopically expressed GFP-IHO1 (**C**) or endogenous IHO1 (**G**) were detected. Bars, 10 μm. **D** Quantification of localization of GFP-tagged IHO1 versions in late zygotene. IHO1 versions: wild type (WT), a mutant missing the last 7 amino acids (C7Δ) and versions where single-letter amino acid code indicates point mutations in positions 569 and 570. Block bars are means. Likelihood-ratio test, ns=$P > 0.05$, ***=$P < 0.001$, ****=$P < 0.0001$. Exact P values: WT vs. C7Δ and AA vs. SA $P < 2.2e\text{-}16$, C7Δ vs. AA $P = 0.06547$, SA vs. SD $P = 9.445e\text{-}10$, SA vs. SE $P = 2.225e\text{-}06$, AA vs. AS $P = 3.217e\text{-}09$, AS vs. DS $P = 4.979e\text{-}14$, AS vs. ES $P = 2.966e\text{-}11$, AA vs. DD $P = 2.599e\text{-}09$, AA vs. EE $P = 0.001003$. **E** Immunoprecipitation (IP) immunoblots from testis extracts of 13 dpp mice. Asterisk and triangles mark unspecific protein band in REC114 blot and isoforms of SYCP3, respectively. Distinct proteins were detected on separate blots. **F** Schematics summarizing conclusions of panel E. See also related Supplementary Fig. 2, Supplementary Table 1 and 2. Source data are provided as a Source Data file.

hinder IHO1-HORMAD1 interaction. However, this interpretation comes with the caveat that there are differences between phosphorylated serines and phosphomimetic amino acids. Hence, the significance of in vivo phosphorylation at S569 and S570 remains uncertain.

To test if disruption of the C-terminus also altered the behavior of endogenous IHO1 we generated a mutant mouse line expressing IHO1_C7Δ from a gene-edited *Iho1* locus (*Iho1^C7Δ/C7Δ^* genotype, Supplementary Fig. 2E). Whereas deletion of the last 7 amino acids of IHO1 did not significantly reduce testicular IHO1 levels (Fig. 1E, Supplementary Fig. 2F), it changed the protein interactions of IHO1, as assayed by immunoprecipitation in testes extracts (Fig. 1E, F, Supplementary Fig. 2G). Wild-type IHO1 formed complexes with HORMAD1, the core axis component SYCP3 and the DSB-factor REC114. In contrast, IHO1_C7Δ efficiently formed complexes with REC114, but not with HORMAD1 or SYCP3 in *Iho1^C7Δ/C7Δ^* mice, which paralleled loss and persistence of IHO1-SYCP3 and IHO1-REC114 complexes, respectively, in *Hormad1^−/−^* mice. These results indicate that the C-terminus of IHO1 enables IHO1-HORMAD1 complex formation, thereby promoting a link between the DSB-machinery and the meiotic chromosome axis.

Importantly, the loss of either HORMAD1 or the last 7 amino acids of IHO1 caused a depletion of IHO1 from chromatin-enriched fractions of testis extracts (Supplementary Fig. 2H). Further, resembling localization of wild-type IHO1 in *Hormad1^−/−^* meiocytes, IHO1_C7Δ did not efficiently accumulate on unsynapsed chromosome axes in either sex of HORMAD1-proficient meiocytes (Fig. 1G, Supplementary Fig. 2I, J). Instead, IHO1_C7Δ formed aggregates, which associated with a few (typically 3-4) chromosome ends in a late zygotene-like stage (Supplementary Fig. 2I). Together, our observations suggest that loss of the last 7 amino acids prevents endogenous IHO1 from efficiently binding HORMAD1 and axes without significantly affecting its other known protein interactions.

In fission yeast, deletion of the C-terminal SSM2/acidic patch of the Mer2/IHO1 ortholog analogously disrupted its interactions with orthologs of both HORMAD1 and the core axis components SYCP2/SYCP3, resulting in its loss from chromosome axes[30]. Thus, the C-terminal SSM2/acidic patch promotes the axis association of Mer2/IHO1-family proteins in distantly related species.

## DDK-activity enhances axial accumulation of IHO1

SDS-PAGE revealed a fast and a slow migrating form of IHO1 (Fig. 2A–C, Supplementary Fig. 2H, 3A and C)[22]. The slow-migrating form was more abundant in chromatin-enriched fractions of testis extracts of wild type (Fig. 2A, Supplementary Fig. 2H), suggesting that IHO1 was posttranslationally modified in correlation with chromatin binding. The slow-migrating form was present in *Hormad1^−/−^* mice (Fig. 2B), where IHO1 was depleted from chromatin (Supplementary Fig. 2H)[22]. Hence, IHO1 modifications may be a cause rather than an outcome of IHO1 chromatin binding. The slow-migrating form was also present in *Spo11^−/−^* mice indicating that it did not require DSB formation (Fig. 2B, Supplementary Fig. 2H). Phosphatase treatment converted the slow- into the fast-migrating form (Fig. 2C, Supplementary Fig. 3A and C),

indicating that the slow-migrating form represented phosphorylated IHO1. IHO1_C7Δ lacks two of the four known phospho-sites of wild-type IHO1. Immunoblot analysis detected only a single IHO1_C7Δ protein band, which had a slightly higher electrophoretic mobility than the fast-migrating form of wild-type IHO1 (Fig. 2A–C). Thus, loss of the last 7 amino acids, and the S569/570 phospho-sites within, may have prevented phosphorylation of all sites in IHO1. Alternatively, IHO1_C7Δ is phosphorylated on S476 and T490 or other unknown sites but without resulting in a strong mobility shift in standard SDS-PAGE. The latter hypothesis was supported by analysis on phos-tag gels (Supplementary Fig. 3B), which enable detection of distinct phosphoforms of proteins by enhancing their retardation during electrophoresis[45,46]. In addition, dephosphorylation induced a slight increase in the electrophoretic mobility of IHO1_C7Δ in standard SDS-PAGE, consistent with residual phosphorylation of unknown sites in IHO1_C7Δ (Supplementary Fig. 3C). Together, these observations suggest that phosphorylation of endogenous IHO1, in particular IHO1 C-terminus, may support IHO1-HORMAD1 interaction and IHO1 recruitment to axes in vivo. IHO1-HORMAD1 interaction may be hindered by phosphomimetic mutations of IHO1 C-terminus (Fig. 1C, D and Supplementary Table 1) because they do not perfectly mimic phophorylation of serines.

S570 is followed by acidic aspartates, characteristic of potential target sites of the DBF4-dependent CDC7 kinase (DDK)[47,48]. DDK is a key activator of the budding yeast IHO1 ortholog, Mer2[49–51], hence DDK may regulate IHO1 in mice. DBF4 interacted with both IHO1 and IHO1_C7Δ in Y2H (Supplementary Fig. 3D). Furthermore, recombinant DDK phosphorylated peptides corresponding to the last 15 amino acids of IHO1 in an in vitro kinase assay; phosphorylation was detected on one but not both serines in about 10% of peptides by mass spectrometry (Fig. 2D, Supplementary Fig. 3E and Supplementary Table 3). The serine that corresponded to S570 of IHO1 was identified as a phospho site with a probability of more than 99.6% by about 50% of annotated spectra. The remaining spectra were annotated with a site probability of 50% for S569 and S570, indicating that only phosphorylation on S570 has been unambiguously identified. These observations indicate that DDK inefficiently phosphorylates S569 in vitro, but DDK may phosphorylate both S569 and S570 in the context of full length IHO1 in vivo.

To address if DDK regulated IHO1 in meiocytes we exposed testis organ cultures to distinct DDK inhibitor cocktails. DDK inhibition impaired IHO1 localization to axes (Fig. 2E, F) and reduced the abundance of the slow-migrating phospho-band of IHO1 in SDS-PAGE (Fig. 2G, H). Further, DDK inhibition lowered recombination focus numbers, as detected by DMC1 staining, in both wild-type and *Iho1^C7Δ/C7Δ^* spermatocytes in early zygotene (Supplementary Fig. 4). These observations suggest that DDK promotes recombination; plausibly through phosphorylation of the IHO1 C-terminus—enhancing axial recruitment of IHO1 by HORMAD1—but also independently of the IHO1 C-terminus.

## Elevated loss of meiocytes in *Iho1^C7Δ/C7Δ^* mice

*Iho1^C7Δ/C7Δ^* male mice did not display infertility (Supplementary Table 4), but they had 1.45 fold smaller testis size (Fig. 3A), and an elevated loss

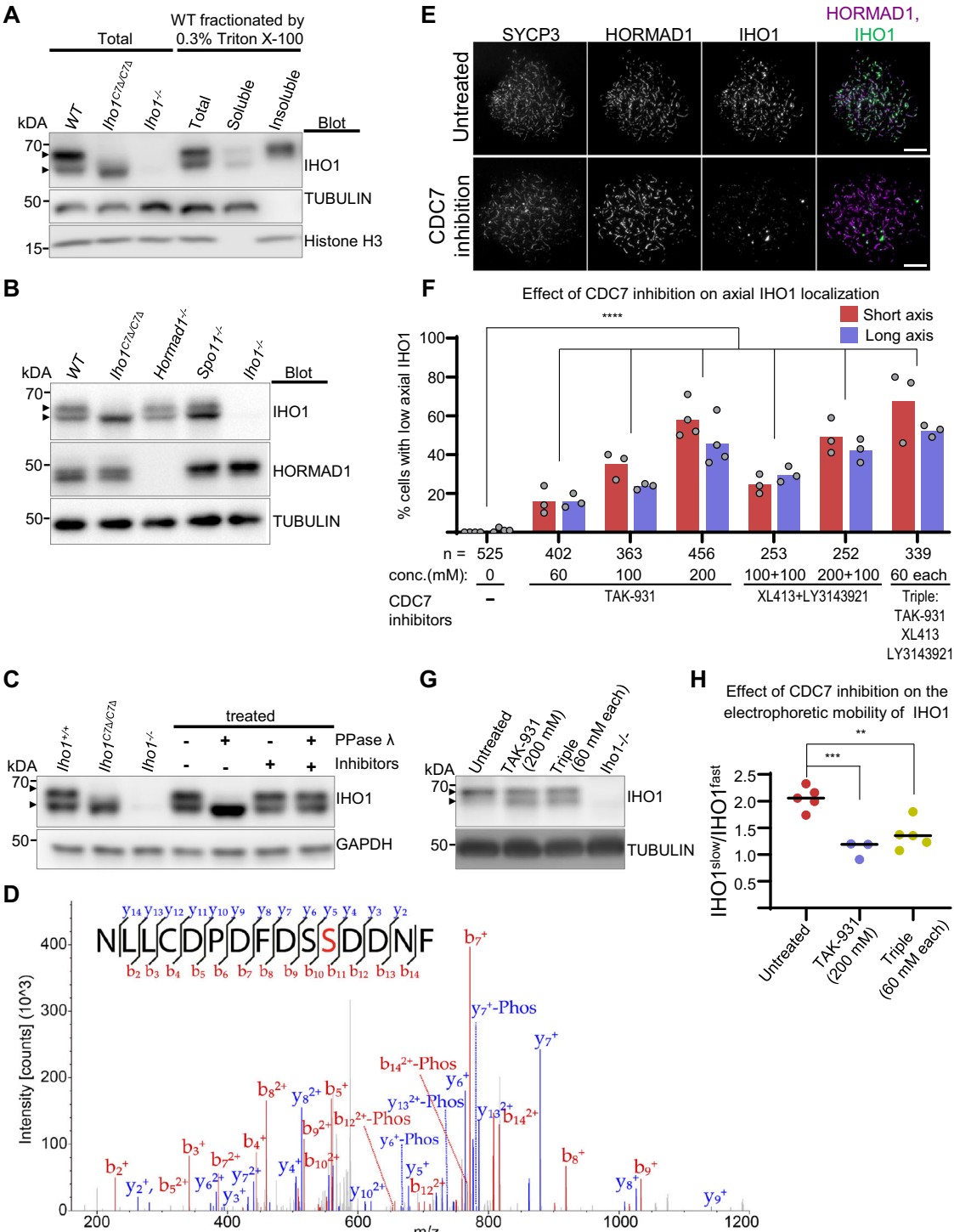

**Fig. 2 | CDC7-dependent phosphoform of IHO1 is enriched in chromatin. A–C** SDS-PAGE immunoblots of protein extracts from testes of 13 dpp mice. Total (**A–C**), fractionated (**A**) and/or phosphatase treated total (**C**) extracts are shown. Distinct proteins were detected on separate blots in **A** and **C**. **D** MS/MS spectrum of NLLCDPDFDS(pS)DNF-COOH peptide. Identified b and y ions are annotated in red and blue, respectively. Fragments with neutral loss of $H_3PO_4$ are indicated as -Phos. 10.9 +/− 1.4% of the peptide was phosphorylated in 2 measurements. **E–H** Analysis of testes of 8 dpp mice after 48 hours in vitro culture with or without CDC7 inhibitors. **E** Nuclear spread spermatocytes. Bars, 10 μm. **F** Quantification of spermatocytes with depleted IHO1 on axes. Block bars are means. Likelihood ratio test,

$P < 2.20E-16$ (****) for comparisons between the untreated sample and all of the CDC7 inhibitor treated samples. **G** SDS-PAGE immunoblots of testis extracts. **A–C**, **G** Arrowheads mark slow and fast migrating IHO1 forms; slow migrating forms were reproducibly more dominant in untreated testis cultures (**G**) than in freshly processed testes (**A–C**). **H** Intensities of slow migrating IHO1 band normalized to intensities of fast migrating bands from SDS-PAGE immunoblots, n = 5 (untreated and triple) or n = 3 (TAK-931 treated) biologically independent samples. Bars are means. Two-tailed t test, \*\*=P = 0.0021, \*\*\*=P = 0.0006. **G–H** Triple inhibitor mix, as in **F**. See also related Supplementary Fig. 3, 4 and Supplementary Table 3. Source data are provided as a Source Data file.

of spermatocytes from mid pachytene stage until meiotic divisions (Supplementary Fig. 5A–C, Supplementary Table 5). Female *Iho1*$^{C7\Delta/C7\Delta}$ mice were also fertile (Supplementary Table 4), but oocyte numbers were approx. 2 fold lower in young *Iho1*$^{C7\Delta/C7\Delta}$ than wild type mice (Supplementary Fig. 5D, E), and fertility of *Iho1*$^{C7\Delta/C7\Delta}$ females declined prematurely (Supplementary Table 4). Meiotic recombination defects trigger elimination of both spermatocytes and oocytes[52–55], therefore *Iho1*$^{C7\Delta/C7\Delta}$ phenotypes were consistent with disrupted meiotic recombination. The phenotypes were milder in *Iho1*$^{C7\Delta/C7\Delta}$ mice as compared to *Hormad1*$^{-/-}$ mice, where inefficient axial recruitment of IHO1 was accompanied by (i) infertility, (ii) a complete elimination of spermatocytes at a mid pachytene-like stage and (iii) a > 3 fold lower testis size as compared to wild type (Fig. 3A)[32,33,56]. Hence, despite similar depletion of IHO1 from axes, meiotic recombination defects are less severe in *Iho1*$^{C7\Delta/C7\Delta}$ than *Hormad1*$^{-/-}$.

### Diminished presence of DSB-factors on chromosome axes in *Iho1*$^{C7\Delta/C7\Delta}$ meiocytes

To further assess the role of the IHO1 C-terminus in recombination, we compared deletions of the IHO1 C-terminus, HORMAD1, or both, for their impact on DSB-factor clusters in spermatocytes (Fig. 3B–G and Supplementary Fig. 6). DSB-factor clusters exist in two types − small clusters (hereafter smDSB-factor clusters) on axes across the genome and large clusters (hereafter laDSB-factor clusters) in mo-2 minisatellite-rich regions, which include PARs and PAR-like telomeric regions of three autosomes[23,24]. IHO1 and HORMAD1 are important for smDSB-factor clusters but dispensable for laDSB-factor clusters[22,24,25,57]. Consistent with the known functions of IHO1[22,24], we observed normal levels of MEI4 and REC114 in *Iho1*$^{C7\Delta/C7\Delta}$ spermatocytes (Supplementary Fig. 6A, B). Furthermore, laDSB-factor clusters (as represented by large, >1 μm diameter MEI4 foci) efficiently assembled at PARs and a subset of autosomal ends in both wild-type and *Iho1*$^{C7\Delta/C7\Delta}$ spermatocytes (Supplementary Fig. 6C, D). In contrast, numbers of smDSB-factor clusters (detected as small REC114-MEI4 co-foci) were strongly reduced (1.8 to 3.9 fold) in *Iho1*$^{C7\Delta/C7\Delta}$ spermatocytes as compared to wild type (Fig. 3B, C). Numbers of smDSB-factor clusters were reduced to similar extents in *Iho1*$^{C7\Delta/C7\Delta}$, *Hormad1*$^{-/-}$ and *Hormad1*$^{-/-}$ *Iho1*$^{C7\Delta/C7\Delta}$ spermatocytes (<1.2 fold differences at matched stages, Fig. 3B, C), suggesting that deletion of IHO1 C-terminus and loss of HORMAD1 impair the same pathway of DSB-machinery assembly.

Although the numbers of smDSB-factor clusters were reduced throughout early prophase, their median signal intensities were significantly higher in *Iho1*$^{C7\Delta/C7\Delta}$, *Hormad1*$^{-/-}$ and *Hormad1*$^{-/-}$ *Iho1*$^{C7\Delta/C7\Delta}$ than in wild-type spermatocytes in the preleptotene stage, when DSBs have not yet formed (Fig. 3D). These signal intensities then gradually dropped below wild-type levels in all three mutants upon progression to early zygotene, when early recombination intermediates reach their peak levels and DSB activity is thought to diminish in wild type. Seeing fewer but brighter clusters at early stages suggests that the seeding of cytologically distinguishable DSB-factor clusters is defective but their initial growth is efficient in the absence of IHO1 C-terminus and/or HORMAD1. DSBs and downstream DNA damage signaling disrupt the DSB-machinery by multiple pathways[34]. Therefore, progressive diminution of smDSB-factor clusters in *Iho1*$^{C7\Delta/C7\Delta}$, *Hormad1*$^{-/-}$ and *Hormad1*$^{-/-}$ *Iho1*$^{C7\Delta/C7\Delta}$ spermatocytes by zygotene might indicate that smDSB-factor clusters are prone to enhanced destabilization by DSBs if IHO1 and HORMAD1 functions are impaired.

To test this hypothesis, we assessed how deletion of the IHO1 C-terminus or absence of HORMAD1 affected smDSB-factor clusters in *Spo11*$^{-/-}$ spermatocytes, where programmed DSBs do not form (Fig. 3E–G). SmDSB-factor clusters were fewer in number but had consistently higher median signal intensities in *Iho1*$^{C7\Delta/C7\Delta}$ *Spo11*$^{-/-}$ and *Hormad1*$^{-/-}$ *Spo11*$^{-/-}$ spermatocytes as compared to *Spo11*$^{-/-}$ spermatocytes in all prophase stages that were present in these genotypes. These observations support the conclusion that smDSB-factor clusters

require HORMAD1 and the IHO1 C-terminus for both efficient seeding and resistance to destabilization by DSBs, but not growth per se.

### The IHO1 C-terminus is required for efficient DSB formation

Given the depletion of smDSB-factor clusters in the absence of the IHO1 C-terminus, we tested if DSB formation was also reduced. Processing of SPO11-generated DSBs results in SPO11-oligo complexes, whose testicular levels inform about DSB activity, allowing comparison of DSB formation in conditions where cellular compositions of testes are similar[58]. Meiotic recombination defects trigger apoptosis in mid pachytene, which is reached at 14 days postpartum (dpp) in the first wave of meiosis in our strain background. Hence, testis cellularity is unaltered by meiotic recombination defects before 14 dpp. Testicular levels of SPO11-oligo complexes were reduced in both *Iho1*$^{C7\Delta/C7\Delta}$ and *Hormad1*$^{-/-}$ as compared to wild type mice at 13 dpp, which indicated that both the IHO1 C-terminus and HORMAD1 were needed for efficient DSB formation (Fig. 4A, B). Deletion of the IHO1 C-terminus did not cause a further reduction of SPO11-oligo levels in a *Hormad1*$^{-/-}$ background (Supplementary Fig. 7A). This supports the hypothesis that loss of the IHO1 C-terminus principally disrupted only those functions of IHO1 in DSB formation that are HORMAD1 dependent.

Processing of DSBs generates single-stranded DNA ends that initiate recombination by invading homologous DNA sequences with the help of recombination proteins that accumulate−manifesting as foci−on single-stranded DNA ends. To further compare the roles of HORMAD1 and IHO1 C-terminus in recombination initiation we examined foci of recombination proteins, DMC1, RAD51 and RPA2, in meiocytes (Fig. 4C–G and Supplementary Fig. 7B-H). *Iho1*$^{C7\Delta/C7\Delta}$ mice showed significantly reduced recombination focus numbers in both spermatocytes (medians, 1.6-2.9 fold in zygotene, Fig. 4F, G, Supplementary Fig. 7B) and oocytes (medians, 2.3-3.1 fold in zygotene, Supplementary Fig. 7F−H) as compared to wild type. Further markers of DSBs, phosphorylated histone H2AX (γH2AX) flares on synapsed autosomes, were also diminished in pachytene stage *Iho1*$^{C7\Delta/C7\Delta}$ spermatocytes as compared to wild type (Supplementary Fig. 7I, J). Together, these observations indicated that DSB activity was reduced in *Iho1*$^{C7\Delta/C7\Delta}$ mice, which likely accounted for defects in meiotic progression and/or fertility.

Interestingly, whereas the low recombination focus numbers were similar in *Hormad1*$^{-/-}$ and *Hormad1*$^{-/-}$ *Iho1*$^{C7\Delta/C7\Delta}$ spermatocytes, focus numbers were consistently lower in these genotypes than in *Iho1*$^{C7\Delta/C7\Delta}$ single mutants (Fig. 4C–G and Supplementary Fig. 7B). The former observation reconfirms that loss of IHO1 C-terminus impairs only HORMAD1-dependent functions of IHO1 in DSB formation. The latter observation may reflect premature DSB repair and/or inappropriate use of sister chromatids instead of homologs as DSB repair templates during meiotic recombination in *Hormad1*$^{-/-}$ but not *Iho1*$^{C7\Delta/C7\Delta}$ meiocytes. Thus, HORMAD1 seems to modulate recombination in addition to recruiting IHO1 to chromosome axes. This conclusion is in agreement with work in budding yeast, where the HORMAD1 ortholog Hop1 not only promotes DSBs, but also enables normal DSB repair kinetics and enhances the use of homologs instead of sister chromatids as recombination partners[59–61]. Previously reported *Hormad1*$^{-/-}$ phenotypes were also consistent with a HORMAD1 role in DSB repair[62,63], and HORMAD1 was hypothesized to prevent premature turnover of early recombination intermediates, thereby sustaining high levels of single-stranded DNA ends for homology search[40,63]. Thus, a conserved HORMAD1 function impacting DSB repair kinetics and/or template choice may explain more severe recombination defects in *Hormad1*$^{-/-}$ than *Iho1*$^{C7\Delta/C7\Delta}$ spermatocytes.

### Assurance of synapsis and CO formation between homologs requires the IHO1 C-terminus

DSBs are needed in high numbers to enable efficient synapsis and crossover formation between each homolog pair in mice[39]. Synapsis is

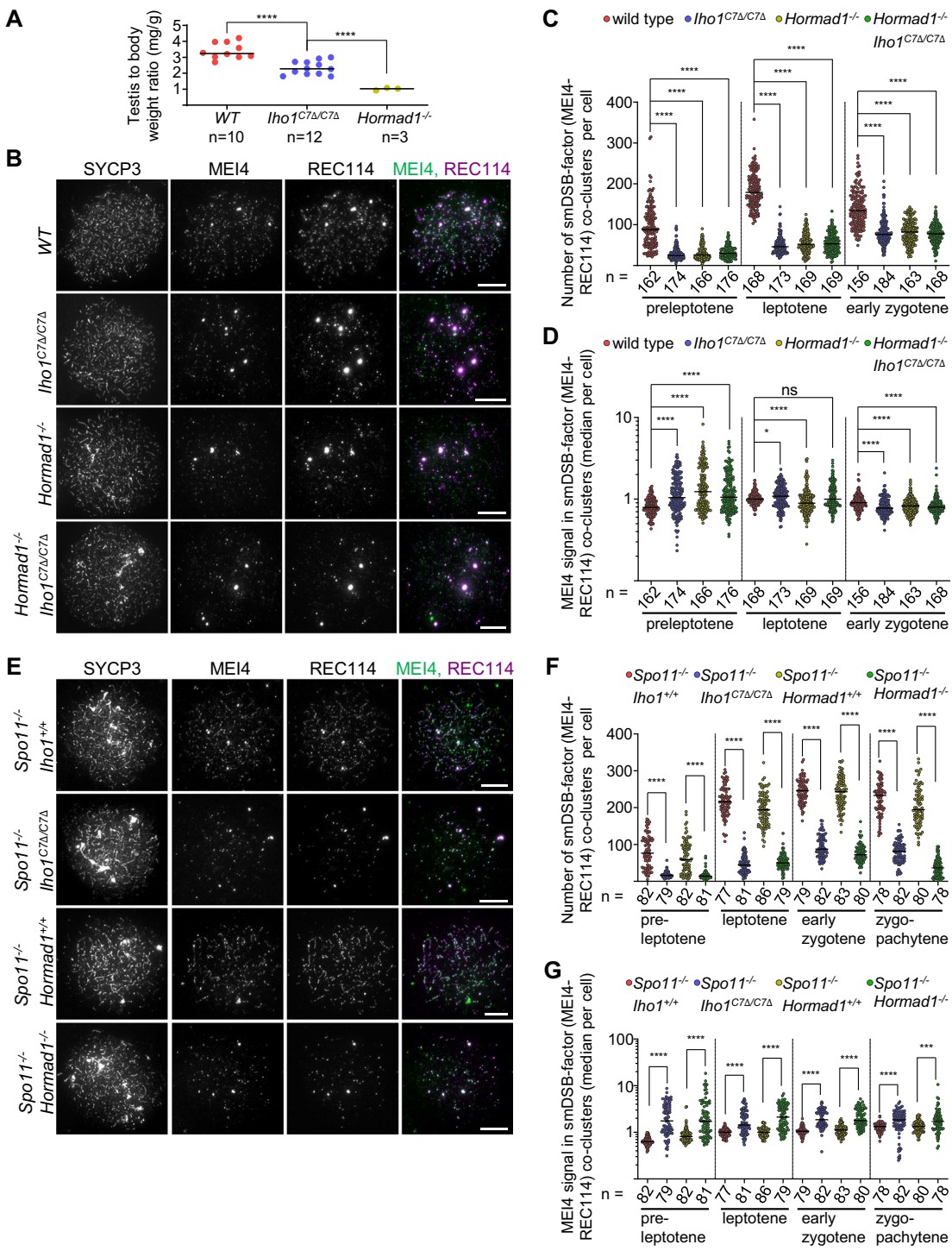

**Fig. 3 | Iho1C7Δ/C7Δ mice show reduction in testis size and DSB-factor cluster numbers. A** Testis to body weight ratios in adult mice (age 50-120 days). Bars mark means. Two-tailed Welch t-test, ****=P < 0.0001, WT vs. *Iho1^C7Δ/C7Δ* P = 4.88e-05, *Iho1^C7Δ/C7Δ* vs. *Hormad1^-/-* P = 1.71e-07. **B, E** Immunostaining in nuclear spread leptotene spermatocytes of adult mice. Bars, 10 μm. **C, D, F, G** Numbers of small MEI4-REC114 co-clusters (**B, E**) and MEI4 intensities in MEI4-REC114 co-clusters (**C, F**, data points show median cluster intensities per cell) in spermatocytes of adult mice. Zygo-pachytene (**F, G**) is equivalent to a mix of late-zygotene and early pachytene stages which are indistinguishable in SC-defective backgrounds. Pooled data is shown from 5 (**C, D**) or 2 (**F, G**) mice of each genotype. Bars are medians, n=cell numbers. Two-tailed Mann Whitney U-Test, ns=P > 0.05, *=P < 0.05, ***=P < 0.001, ****=P < 0.0001. Exact P values: (**C, F**), P < 2.2e-16 for all comparisons, (**D**), pre-leptotene, wild type vs. *Iho1^C7Δ/C7Δ*, P = 1.63e-10, wild type vs. *Hormad1^-/-*, P = 3.39e-15, wild type vs. *Iho1^C7Δ/C7Δ Hormad1^-/-*, P = 1.97e-11, leptotene, wild type vs. *Iho1^C7Δ/C7Δ*,

P = 0.04839, wild type vs. *Hormad1^-/-*, P = 3.57e-5, wild type vs. *Iho1^C7Δ/C7Δ Hormad1^-/-*, P = 0.782, early zygotene, wild type vs. *Iho1^C7Δ/C7Δ*, P = 1.84e-7, wild type vs. *Hormad1^-/-*, P = 1.53e-5, wild type vs. *Iho1^C7Δ/C7Δ Hormad1^-/-*, P = 3.42e-5, (**G**), pre-leptotene, *Spo11^-/- Iho1^+/+* vs. *Spo11^-/- Iho1^C7Δ/C7Δ*, P < 2.2e-16, *Spo11^-/- Hormad1^+/+* vs. *Spo11^-/- Hormad1^-/-* P = 5.03e-11, leptotene, *Spo11^-/- Iho1^+/+* vs. *Spo11^-/- Iho1^C7Δ/C7Δ*, P = 1.37e-10, *Spo11^-/- Hormad1^+/+* vs. *Spo11^-/- Hormad1^-/-* P = 9.16e-15, early zygotene, *Spo11^-/- Iho1^+/+* vs. *Spo11^-/- Iho1^C7Δ/C7Δ*, P < 2.2e-16, *Spo11^-/- Hormad1^+/+* vs. *Spo11^-/- Hormad1^-/-* P < 2.2e-16, zygo-pachytene, *Spo11^-/- Iho1^+/+* vs. *Spo11^-/- Iho1^C7Δ/C7Δ*, P = 8.4e-7, *Spo11^-/- Hormad1^+/+* vs. *Spo11^-/- Hormad1^-/-* P = 0.0003408. Statistical tests compare samples that were stained and processed in parallel within experimental repeats to reduce technical variability. Thus, *Spo11^-/- Iho1^+/+and C7Δ/C7Δ* are not directly comparable with *Spo11^-/- Hormad1^+/+and-/-* due to sample preparation from different colonies on different days (**F, G**). See also related Supplementary Fig. 5, 6 and Supplementary Tables 4 and 5. Source data are provided as a Source Data file.

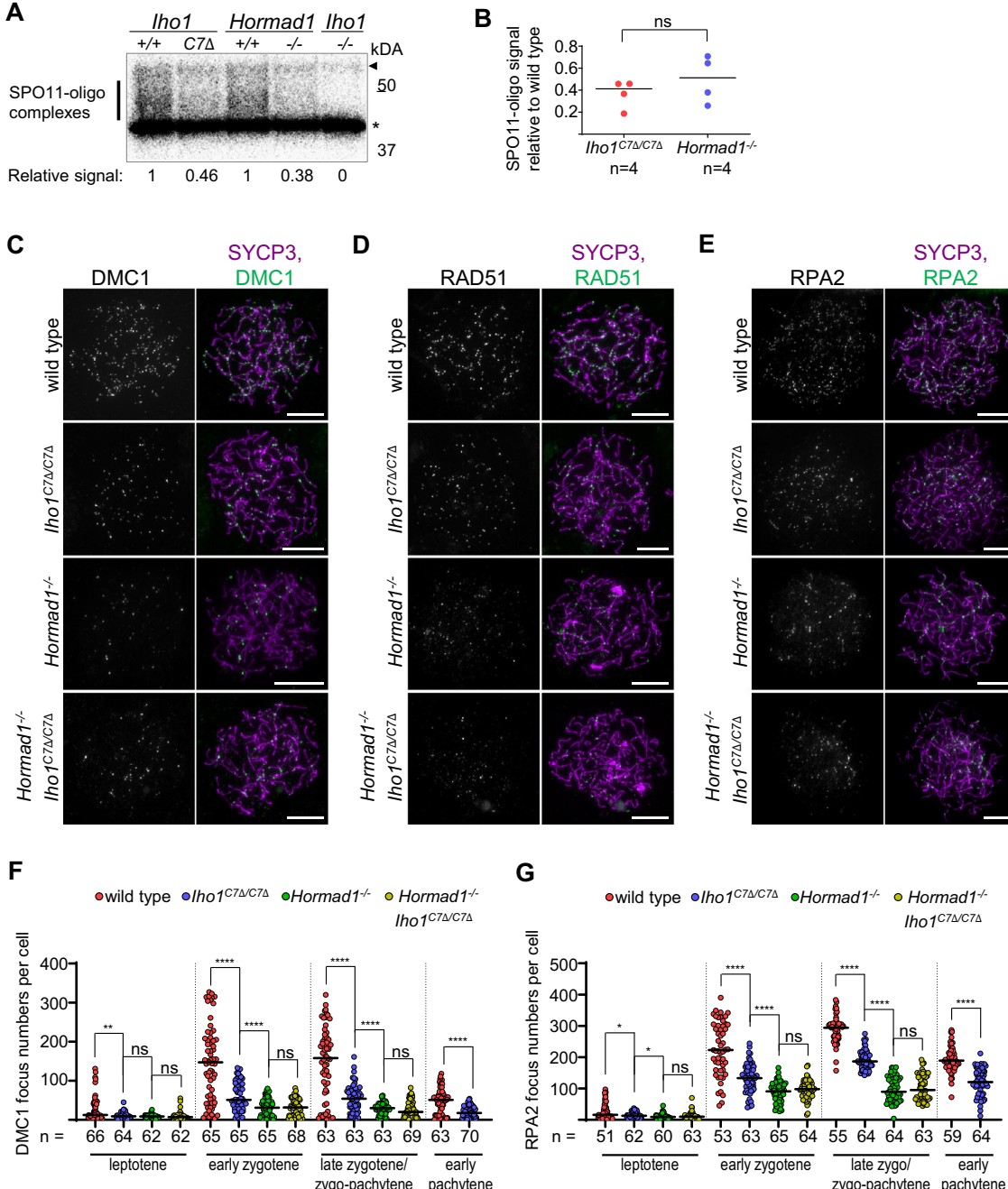

**Fig. 4 | Efficient meiotic recombination relies on IHO1-HORMAD1 interaction.**
**A** Radiograph of immunoprecipitated and radioactively labeled SPO11-oligo complexes from testes of 13 dpp juvenile mice. Bar, SPO11-specific signals, asterisk, nonspecific labelling, and arrowhead, immunoglobulin heavy-chain. Radioactive signals were background-corrected (*Iho1*⁻/⁻, signal=0) and normalized to corresponding wild type control (1). **B** Quantification of SPO11-oligo complexes from 13dpp mice. Bars are mean, n=number of mice. Two-tailed paired t-test, ns=P = 0.3419. **C–E** Immunostaining in nuclear spread early zygotene spermatocytes of adult mice. Bars, 10 μm. **F, G** Quantification of axis associated DMC1 (**F**), RPA2 (**G**) focus numbers in spermatocytes. Pools of two experiments are shown (one mouse represented each genotype in each experiment). Bars are medians, n=cell numbers. Two-tailed Mann Whitney U-test, ns=P > 0.05, *=P < 0.05, **=P < 0.01, ****=P < 0.0001. Exact P values: (**F**), leptotene, wild type vs. *Iho1*^{C7Δ/C7Δ}, P = 0.05,

*Iho1*^{C7Δ/C7Δ} vs. *Hormad1*⁻/⁻, P = 0.486, *Hormad1*⁻/⁻ vs. *Hormad1*⁻/⁻ *Iho1*^{C7Δ/C7Δ}, P = 0.234, early zygotene, wild type vs. *Iho1*^{C7Δ/C7Δ}, P = 1.26e-7, *Iho1*^{C7Δ/C7Δ} vs. *Hormad1*⁻/⁻, P = 7.38e-6, *Hormad1*⁻/⁻ vs. *Hormad1*⁻/⁻ *Iho1*^{C7Δ/C7Δ}, P = 0.984, late zygotene, wild type vs. *Iho1*^{C7Δ/C7Δ}, P = 9.44e-9, *Iho1*^{C7Δ/C7Δ} vs. *Hormad1*⁻/⁻, P = 1.61e-6, *Hormad1*⁻/⁻ vs. *Hormad1*⁻/⁻ *Iho1*^{C7Δ/C7Δ}, P = 0.373, pachytene, wild type vs. *Iho1*^{C7Δ/C7Δ}, P = 1.5e-6, (**G**), leptotene, wild type vs. *Iho1*^{C7Δ/C7Δ}, P = 0.02417, *Iho1*^{C7Δ/C7Δ} vs. *Hormad1*⁻/⁻, P = 0.01189, *Hormad1*⁻/⁻ vs. *Hormad1*⁻/⁻ *Iho1*^{C7Δ/C7Δ}, P = 0.2825, early zygotene, wild type vs. *Iho1*^{C7Δ/C7Δ}, P = 2.19e-9, *Iho1*^{C7Δ/C7Δ} vs. *Hormad1*⁻/⁻, P = 3.17e-8, *Hormad1*⁻/⁻ vs. *Hormad1*⁻/⁻ *Iho1*^{C7Δ/C7Δ}, P = 0.2455, late zygotene, wild type vs. *Iho1*^{C7Δ/C7Δ}, P < 2.2e-16, *Iho1*^{C7Δ/C7Δ} vs. *Hormad1*⁻/⁻, P < 2.2e-16, *Hormad1*⁻/⁻ vs. *Hormad1*⁻/⁻ *Iho1*^{C7Δ/C7Δ}, P = 0.4054, pachytene, wild type vs. *Iho1*^{C7Δ/C7Δ}, P = 7.28e-15. See also related Supplementary Fig. 7. Source data are provided as a Source Data file.

not completed in most spermatocytes if DSB activity is below 50% of wild-type levels according to analysis of a *Spo11*-hypomorphic mutant mouse line, *Tg(Spo11β)*^{+/−39}. Approximately 2 fold fewer DSBs were found in *Iho1*^{C7Δ/C7Δ} as compared to wild type (Fig. 4 and Supplementary

Fig. 7A–H), hence we tested if reduced DSB activity resulted in downstream synapsis errors in *Iho1*^{C7Δ/C7Δ} meiocytes.

A subset of *Iho1*^{C7Δ/C7Δ} meiocytes formed synaptonemal complexes (SCs) on all chromosomes, but efficiency and accuracy of SC

formation was reduced in $Iho1^{C7\Delta/C7\Delta}$ as compared to wild type in both sexes (Fig. 5A–D and Supplementary Fig. 8 and 9A–E). SCs connected PARs of sex chromosomes and the full length of autosomes in most wild-type spermatocytes in pachytene stage. In contrast, pachytene-like $Iho1^{C7\Delta/C7\Delta}$ spermatocytes featured four main types of synapsis defects, (i) non-homologous entanglements/ synapsis, (ii) partially unsynapsed autosomes, (iii) fully unsynapsed PARs and/or (iv) autosomes (Fig. 5B, C and Supplementary Fig. 9A). Whereas frequencies of non-homologous synapsis and autosomal asynapsis decreased as pachytene progressed, sex-chromosome asynapsis grew more frequent (Fig. 5B). In most of the asynaptic $Iho1^{C7\Delta/C7\Delta}$ spermatocytes ($n = 175$), one to two autosomes were affected (average 1.12).

Ranking of chromosome axis length suggested that autosomes of all lengths were prone to asynapsis in early pachytene (Fig. 5C). In contrast, asynapsis was more frequent for short chromosomes in late pachytene. These observations suggest that, in contrast to short asynaptic chromosomes, long asynaptic chromosomes eventually synapse during pachytene in $Iho1^{C7\Delta/C7\Delta}$ spermatocytes. Alternatively, asynapsis of long chromosomes may be more likely to trigger apoptosis, consistent with prior observations that only extensive asynapsis can do so[53]. Synapsis defects originating from reduced DSB numbers likely underlie both the low testis size (Fig. 3A) and low oocyte numbers (Supplementary Fig. 5D, E) in $Iho1^{C7\Delta/C7\Delta}$ mice, as asynapsis triggers apoptosis in both spermatocytes and oocytes[52,54,55,64].

We further tested the formation of crossover-specific recombination intermediates, represented by foci of MLH1, a component of the Holliday junction resolvase[65–68]. A large fraction of $Iho1^{C7\Delta/C7\Delta}$ oocytes (40%, $n = 68$) and spermatocytes (35%, $n = 100$) formed MLH1 foci on all of their chromosomes in pachytene (Fig. 5E and Supplementary Fig. 9F). Nonetheless, defects in MLH1 focus formation were apparent (Fig. 5E–G and Supplementary Fig. 9F–H). $Iho1^{C7\Delta/C7\Delta}$ meiocytes always lacked MLH1 foci on chromosomes that were fully unsynapsed (12 autosomes and 25 sex chromosome pairs in 100 spermatocytes, 32 chromosomes in 68 oocytes), and frequently lacked MLH1 foci on partially unsynapsed chromosomes (24 out of 61 chromosomes, quantified only in oocytes (the same oocyte population as above), where asynapsis was frequent (see asynapsis frequency in Supplementary Fig. 9D). Interestingly, small fractions of fully synapsed chromosomes of $Iho1^{C7\Delta/C7\Delta}$ meiocytes also lacked MLH1 foci in both sexes (3% in oocytes, Supplementary Fig. 9G, 0.94% autosomes in spermatocytes, Fig. 5F). In spermatocytes, MLH1 focus formation was affected not only on autosomes, but also on sex chromosomes that synapsed their PARs − MLH1 was missing from 28% of PARs in wild type ($n = 85$) and 53% in $Iho1^{C7\Delta/C7\Delta}$ ($n = 74$; Fisher exact test, $P = 0.002$), respectively. Accordingly, total MLH1 focus numbers were slightly lower in $Iho1^{C7\Delta/C7\Delta}$ than wild-type mice when meiocytes with fully synapsed chromosomes were considered (2 and 1 foci lower medians in males and females, respectively; Fig. 5G and Supplementary Fig. 9H).

One possible interpretation is that the reduced DSB numbers in $Iho1^{C7\Delta/C7\Delta}$ meiocytes exceeds the capacity of the as-yet enigmatic crossover homeostasis mechanism, which keeps crossover numbers stable even if numbers of recombination initiation events vary[69,70]. Alternatively, the IHO1 C-terminus may support post-DSB functions of IHO1 that promote crossover formation, analogous to functions reported for Mer2 in the fungus *Sordaria macrospora* (*Sm*)[14]. We favor the former hypothesis because − contrary to *Sm*Mer2, which localizes to SC central regions and CO-specific recombination foci following chromosomal synapsis[14] − IHO1 is not detectable on synapsed chromosomal regions[22,34], where CO-generating recombination occurs. Further, conditional depletion of IHO1 soon after most DSBs formed in zygotene does not cause obvious defects in meiosis in mice[34].

Therefore, post-DSB functions of *Sm*Mer2 are unlikely to be conserved in mammals.

## IHO1_C7Δ does not support accumulation of recombination initiation events in unsynapsed regions

SC formation was suggested to restrict DSB activity to late synapsing genomic regions, where newly formed DSBs are useful for completion of homolog synapsis[8,20,22,39,40]. Confirming this hypothesis, delayed synapsis provokes persistence of DSB-machinery[34,35,37] and accumulation of DSBs[37–39] in affected genomic regions in both budding yeast and mice. Asynapsis was frequently observed in *Spo11*-hypomorphic *Tg(Spo11β)*[+/−] spermatocytes, due to an over 2 fold reduction in total DSB activity as compared to wild type[39]. Curiously, despite the overall low DSB levels, DSB density reached wild type levels on unsynapsed axes as *Tg(Spo11β)*[+/−] spermatocytes progressed to advanced-late zygotene, where less than 30% of axes were unsynapsed. This phenomenon was attributed to persistent DSB activity in regions where synapsis was delayed.

If, similar to *Tg(Spo11β)*[+/−], $Iho1^{C7\Delta/C7\Delta}$ spermatocytes were able to top up DSBs on asynaptic axes, the relative density of DSB markers on unsynapsed versus synapsed axis (hereafter unsynapsed-to-synapsed DSB density ratio) would be the same or higher in $Iho1^{C7\Delta/C7\Delta}$ as compared to wild-type in an advanced-late zygotene stage. However, our observations suggested that DSB-factor clusters were not only inefficiently seeded, but were also sensitive to negative-feedback signaling from DSBs in $Iho1^{C7\Delta/C7\Delta}$ meiocytes (Fig. 3). Hence, beyond having overall low DSB activity, $Iho1^{C7\Delta/C7\Delta}$ meiocytes might fail to efficiently maintain DSB activity on asynaptic axes once DSBs form in leptotene and zygotene.

To test this hypothesis, we examined densities of DSBs on axes by co-staining DMC1 and RPA2, which are preferential markers of recombination foci in unsynapsed and synapsed regions, respectively (Fig. 6A). We compared $Iho1^{C7\Delta/C7\Delta}$ with wild type and two DSB-defective genotypes, *Spo11*[+/−] and *Ankrd31*[−/−]. In *Spo11*[+/−] spermatocytes, DSB formation was mildly delayed and reduced (15-30%) relative to wild type[69]. In *Ankrd31*[−/−] spermatocytes, both DSB-factor cluster formation and recombination initiation were strongly delayed, but DSB focus numbers reached or surpassed wild type levels in late zygotene[23,24]. Unsynapsed-to-synapsed DSB density ratios were below one in $Iho1^{C7\Delta/C7\Delta}$ spermatocytes (median=0.83) and above 1 in wild-type (median=1.1), *Spo11*[+/−] (median=1.17,) and *Ankrd31*[−/−] (median=1.58) spermatocytes in advanced-late zygotene (Fig. 6B).

These observations suggest that late-synapsing regions in $Iho1^{C7\Delta/C7\Delta}$ spermatocytes have comparatively low DSB activity. Further, asynaptic regions cannot top up DSBs to levels of early synapsing regions, unlike wild-type, *Spo11*[+/−] and *Ankrd31*[−/−] spermatocytes. Consistent with these conclusions, DSB densities were substantially lower on unsynapsed axes of $Iho1^{C7\Delta/C7\Delta}$ than either wild type (~2.4 fold), *Spo11*[+/−] (~2.2 fold) or *Ankrd31*[−/−] (~3.7 fold) spermatocytes in advanced-late zygotene (Fig. 6C–G). Spreading of SC from 20-70% to over 70% of total axis length was not accompanied by a significant increase in DSB densities on unsynapsed axes of late zygotene wild-type or $Iho1^{C7\Delta/C7\Delta}$ spermatocytes (Fig. 6C–E). In contrast, a small but significant increase of DSB densities was observed on unsynapsed axes of *Spo11*[+/−] and *Ankrd31*[−/−] spermatocytes as synapsis progressed (Fig. 6C, F, G), which was in line with delayed DSB kinetics in *Spo11*[+/−] and *Ankrd31*[−/−] mice[23,24,69].

Steadily low DSB densities in $Iho1^{C7\Delta/C7\Delta}$ spermatocytes suggest that asynapsis does not enable enduring DSB formation in the absence of IHO1-HORMAD1 interaction. Further, high DSB densities on unsynapsed axes in late zygotene *Ankrd31*[−/−] spermatocytes suggest that, whereas both ANKRD31 and IHO1-HORMAD1 promote DSB-factor clusters on axes, only IHO1-HORMAD1 is required for maintaining DSB activity if synapsis is delayed.

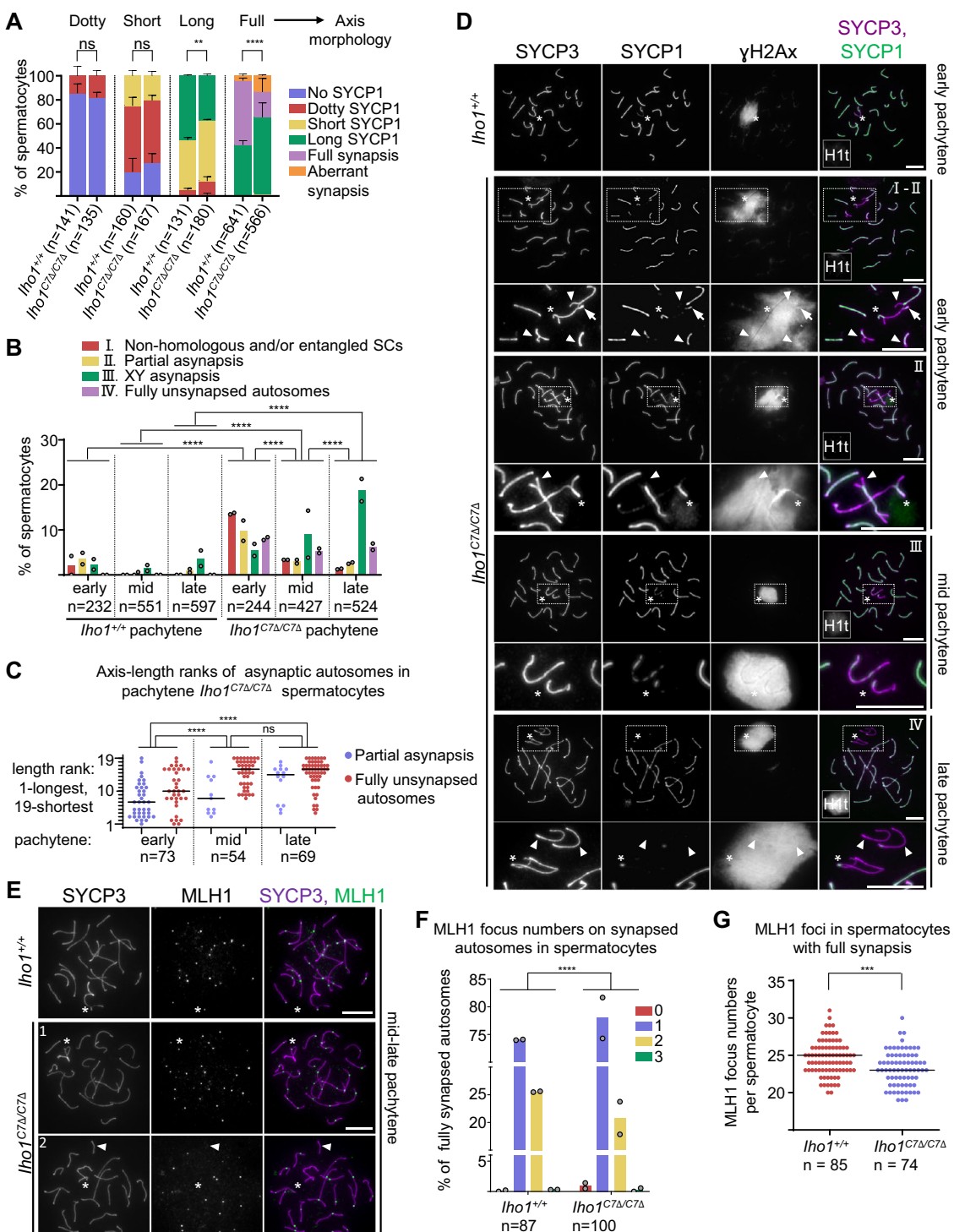

**Fig. 5 | Efficient SC and CO formation require IHO1-HORMAD1 interaction. A-C, F, G** Quantifications of SC morphology relative to axis development (**A**), I-IV non-exclusive SC defect types (**B**), axis-length ranks of partially (blue) or completely (red) unsynapsed autosomes (**C**), MLH1 focus numbers on synapsed autosomes (**F**) or total MLH1 numbers (**G**) in spermatocytes of adult mice. Leptotene to early pachytene cells (identified by a lack of histone H1t) (**A**), pachytene cells (**B**, **C**), or both fully synapsed and asynaptic (**F**) or only fully synapsed (**G**) late pachytene cells were examined. Pools of two experiments are shown (one mouse represented each genotype in each experiment). Bars are means (**A**, **B**, **F**) or medians (**C**, **G**), error bars are standard deviation (**A**), n=numbers of spermatocytes. Likelihood ratio test (**A**, **B**, **F**) or two-tailed Mann Whitney U-Test (**C**, **G**), ns=*P* > 0.05, **=*P* < 0.01, ***=*P* < 0.001, ****=*P* < 0.0001. Exact P values: (**A**), (dotty axis) P = 0.5195, (short axis) *P* = 0.08376, (long axis) *P* = 0.00735, (full axis) *P* < 2.2e-16, (**B**), *Iho1*[+/+] vs. *Iho1*[C7Δ/C7Δ]

(early pachytene) *P* = 1.1e-9, (mid-pachytene) *P* = 1.97e-14, (late-pachytene) *P* < 2.2e-16, *Iho1*[C7Δ/C7Δ] early vs. mid-pachytene *P* = 6.43e-7, *Iho1*[C7Δ/C7Δ] mid- vs. late pachytene *P* = 1.7e-5, (**C**), early vs. mid- *P* = 6.26e-3, early vs. late *P* = 8.53e-8, mid- vs. late P = 0.634, (**F**), P = 0.0001898, (**G**) *P* = 5.51e-5. **D**, **E** Immunostaining in nuclear spread spermatocytes of adult mice. Asterisks, sex chromosomes. **D** Insets are enlarged under respective panels. Arrowheads mark partially or fully unsynapsed autosomes, arrow marks non-homologous SC. Miniaturized images (left bottom corners) show histone H1t, a marker of mid and post-mid pachytene stages (intermediate or high levels, respectively). Roman numbers refer to SC defect types as described in **B**. **E** *Iho1*[C7Δ/C7Δ] cells where either all chromosomes have MLH1 (cell 1) or a synapsed autosome lacks MLH1 (cell 2, arrowhead). **D**, **E** Bars, 10 μm. See also related Supplementary Fig. 9. Source data are provided as a Source Data file.

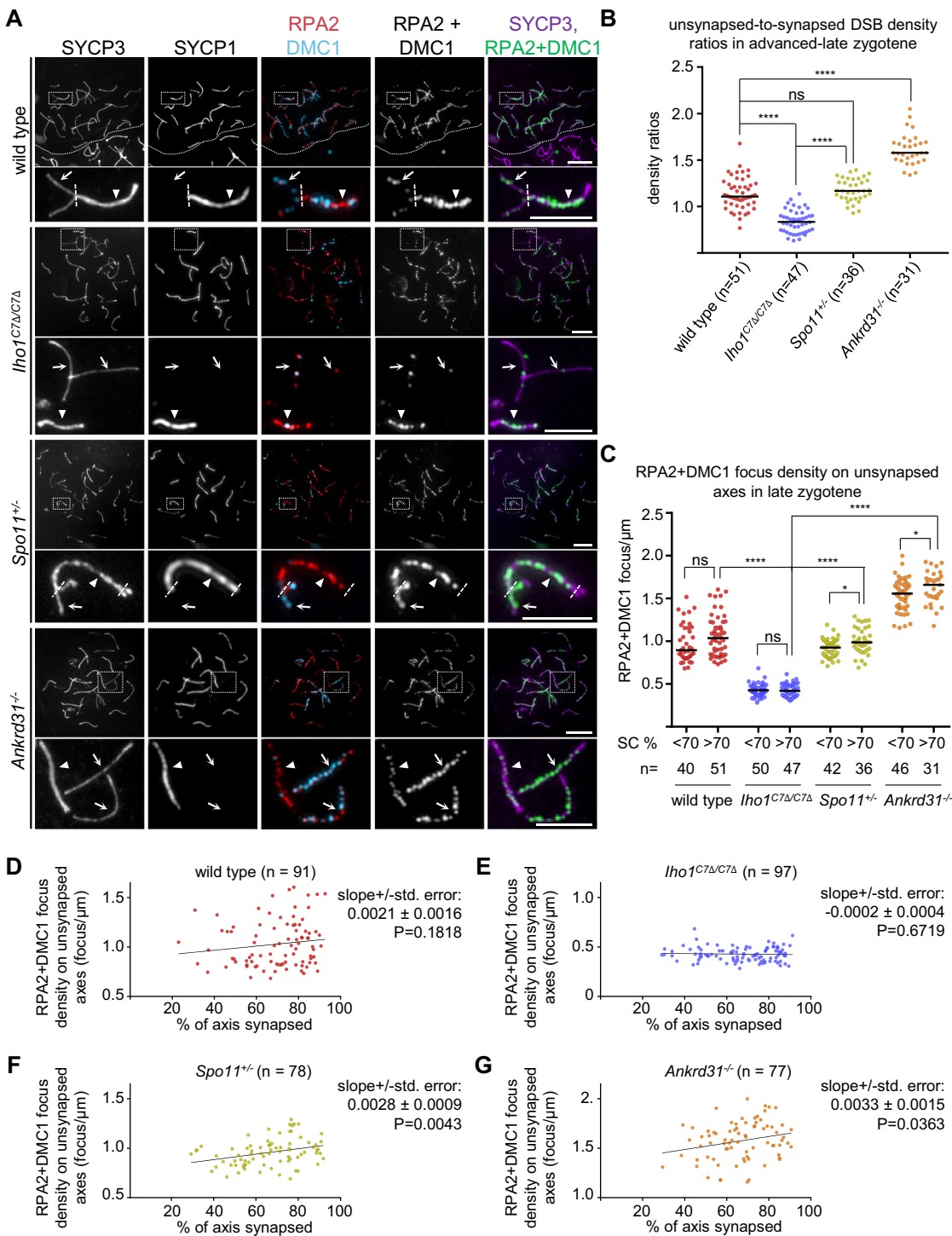

**Fig. 6 | Disruption of IHO1-HORMAD1 interface prevents DSB top-up in late synapsing axes. A** Immunostaining in nuclear spread late zygotene spermatocytes of adult mice. In wild type, dotted line separates a zygotene (top) from a diplotene (bottom) cell. In enlarged insets, dashed lines mark borders between unsynapsed (arrow) and synapsed (triangle) axes. Bars, 10 and 5 μm in low and high magnification panels, respectively. **B** Unsynapsed-to-synapsed DMC1 + RPA2 focus density ratios in late zygotene cells where SC formed on >70% of total axis length. **C–G** DSB focus densities on unsynapsed axes in late zygotene cells grouped (**C**, SC formed on <70% or >70% of total axis length) or ordered (**D–G**) according to the extent of synapsis nucleus-wide. **B–G** Each dot represents a single cell. n=numbers of

spermatocytes. **B, C** Bars, medians, two-tailed Mann Whitney U-Test, ns=$P > 0.05$, *=$P < 0.05$, ****=$P < 0.0001$. Exact P values: (**B**), wild type vs. $Iho1^{C7\Delta/C7\Delta}$ $P = 1.27e$-10, wild type vs. $Spo11^{-/-}$ $P = 0.1198$, wild type vs. $Ankrd31^{-/-}$ $P = 5.99e$-10, $Iho1^{C7\Delta/C7\Delta}$ vs. $Spo11^{-/-}$ $P < 2.2e$-16, (**C**), comparison of >70% and <70% synapsed spermatocytes in wild type $P = 0.06574$, $Iho1^{C7\Delta/C7\Delta}$ $P = 0.7645$, $Spo11^{-/-}$ $P = 0.01217$, $Ankrd31^{-/-}$ $P = 0.04398$; comparison of spermatocytes with >70% synapsis, wild type vs. $Iho1^{C7\Delta/C7\Delta}$ $P < 2.2e$-16, $Iho1^{C7\Delta/C7\Delta}$ vs. $Spo11^{-/-}$ $P < 2.2e$-16, $Iho1^{C7\Delta/C7\Delta}$ vs. $Ankrd31^{-/-}$ $P < 2.2e$-16. **D–G** Linear regression (lines), the best-fit slope +/− standard error and the significance of slope deviation from zero (two-tailed F test, P) are shown. Source data are provided as a Source Data file.

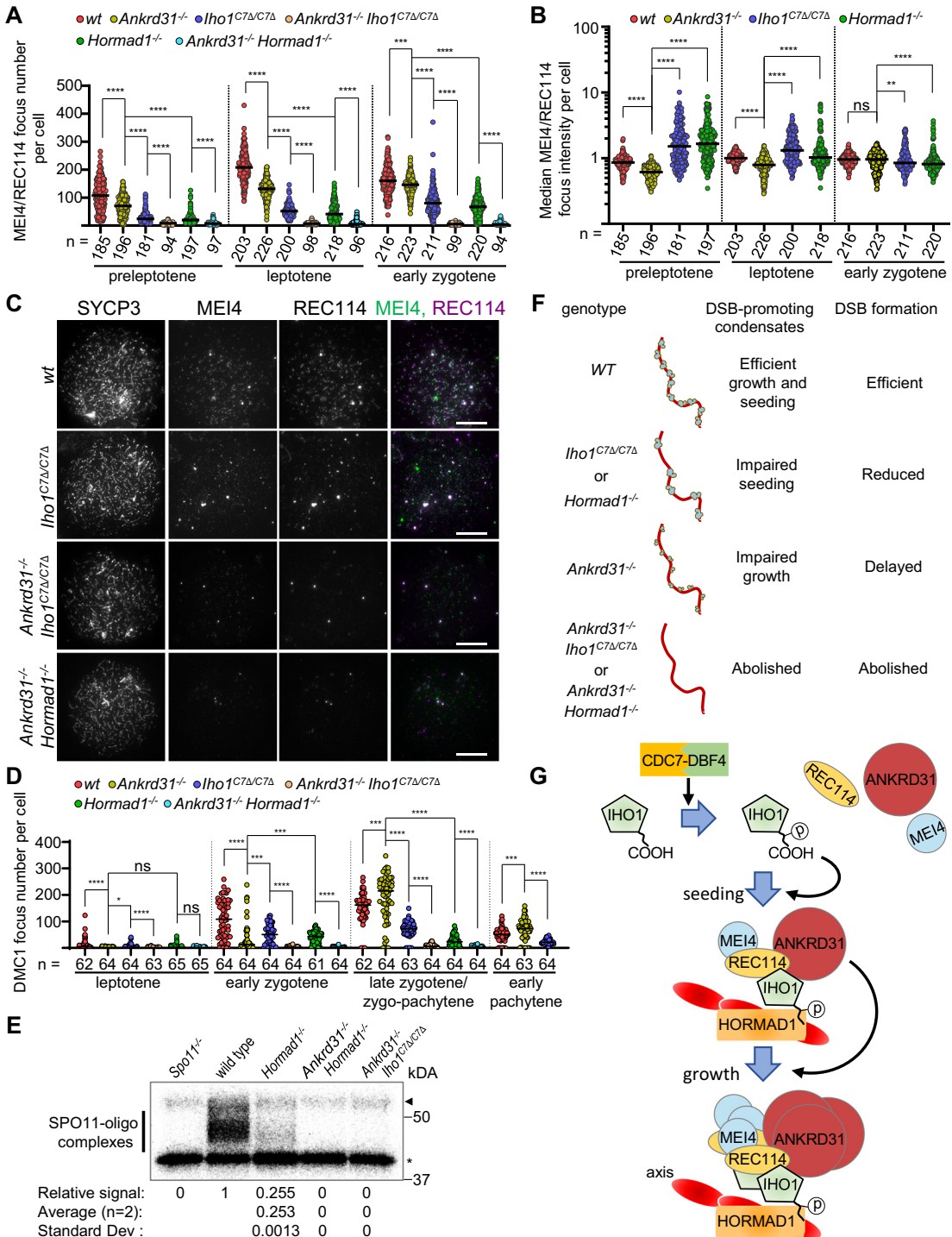

### The IHO1-HORMAD1 interaction and ANKRD31 redundantly enable DSB activity

Not only DSB dynamics (Fig. 6), but also DSB-factor clusters were differentially affected by disruptions of ANKRD31 or IHO1-HORMAD1 ([23–25] and Fig. 7A, B). Whereas ANKRD31 loss led to disappearance of laDSB-factor clusters in PARs and PAR-like autosomal telomeres[23,24], loss of IHO1[24], HORMAD1[25] or the IHO1 C-terminus (Supplementary Fig. 6C, D) had no or very little effect on laDSB-factor clusters. Further, initial growth of smDSB-factor clusters was impaired in the absence of ANKRD31, but not in the absence of IHO1 C-terminus or HORMAD1 (Figs. 3E, G, 7B).

ANKRD31 directly interacts with IHO1, REC114 and MEI1 by partially distinct domains ([24], Supplementary Fig. 10A and Supplementary

Table 6). Therefore, ANKRD31 may enhance DSB-factor clustering by increasing interconnectivity between cluster components, contrasting a proposed IHO1-HORMAD1 function in anchoring DSB-factor clusters to axes. Accordingly, ANKRD31 and IHO1-HORMAD1 may promote DSB-factor clustering by redundant pathways.

DSB-factor clusters were detected in negligible numbers if both ANKRD31 and either the IHO1 C-terminus or HORMAD1 were disrupted (Fig. 7A, C and Supplementary Fig. 10B). DSB-factor clusters resembling PAR-linked clusters of wild type were still present on chromatin of *Iho1^C7Δ/C7Δ Ankrd31^-/-* (92.7%, n = 193 cells) and *Hormad1^-/- Ankrd31^-/-* (91.2%, n = 136 cells, Fig. 7C and Supplementary Fig. 10B) from preleptotene to early zygotene. However, these clusters rarely associated with PAR-FISH signals (11.4%, n = 603 clusters in *Iho1^C7Δ/C7Δ Ankrd31^-/-*,

**Fig. 7 | IHO1-HORMAD1 complex and ANKRD31 are redundantly necessary for meiotic DSBs. A, B, D** Numbers of small MEI4-REC114 co-clusters (**A**), MEI4 intensities in MEI4-REC114 co-clusters (**B**, data points show median cluster intensities per cell), and DMC1 focus numbers (**D**) in spermatocytes of adult mice. Zygo-pachytene (**D**), only in SC-defective backgrounds) is equivalent to a mix of late-zygotene and early pachytene stages which are indistinguishable if SC is defective. Pooled data is shown from 6 (**A, B**) or 2 (**D**) mice of each genotype. Bars are medians, n=cell numbers. Two-tailed Mann Whitney U-Test, ns=$P > 0.05$, *=$P < 0.05$, **=$P < 0.01$, ***=$P < 0.001$, ****=$P < 0.0001$. Exact P values: (**A**), wild type (wt) versus *Ankrd31*$^{-/-}$ pre-leptotene $P = 5.35e-13$, early zygotene $P = 0.0001596$, all the others $P < 2.2e-16$, (**B**), all comparisons in pre-leptotene and leptotene $P < 2.2e-16$, early zygotene, wt vs. *Ankrd31*$^{-/-}$ $P = 0.3047$, *Ankrd31*$^{-/-}$ vs. *Iho1*$^{C7\Delta/C7\Delta}$ $P = 0.001186$, *Ankrd31*$^{-/-}$ vs. *Hormad1*$^{-/-}$ $P = 7.19e-5$, (**D**), leptotene, wt vs. *Ankrd31*$^{-/-}$ $P = 1.07e-6$, *Ankrd31*$^{-/-}$ vs. *Iho1*$^{C7\Delta/C7\Delta}$ $P = 0.01621$, *Iho1*$^{C7\Delta/C7\Delta}$ vs. *Ankrd31*$^{-/-}$ *Iho1*$^{C7\Delta/C7\Delta}$ $P = 5.28e-5$, *Ankrd31*$^{-/-}$ vs. *Hormad1*$^{-/-}$ $P = 0.1945$, *Hormad1*$^{-/-}$ vs. *Ankrd31*$^{-/-}$ *Hormad1*$^{-/-}$ $P = 0.175$, early zygotene, wt vs. *Ankrd31*$^{-/-}$ $P = 5.82e-10$, *Ankrd31*$^{-/-}$ vs. *Iho1*$^{C7\Delta/C7\Delta}$ $P = 0.00041$, *Iho1*$^{C7\Delta/C7\Delta}$ vs. *Ankrd31*$^{-/-}$ *Iho1*$^{C7\Delta/C7\Delta}$ $P = 1.47e-13$, *Ankrd31*$^{-/-}$ vs. *Hormad1*$^{-/-}$

$P = 0.0002882$, *Hormad1*$^{-/-}$ vs. *Ankrd31*$^{-/-}$ *Hormad1*$^{-/-}$ $P < 2.2e-16$, late zygotene, wt vs. *Ankrd31*$^{-/-}$ $P = 0.0004372$, *Ankrd31*$^{-/-}$ vs. *Iho1*$^{C7\Delta/C7\Delta}$ $P = 1.48e-13$, *Iho1*$^{C7\Delta/C7\Delta}$ vs. *Ankrd31*$^{-/-}$ *Iho1*$^{C7\Delta/C7\Delta}$ $P = 5.35e-16$, *Ankrd31*$^{-/-}$ vs. *Hormad1*$^{-/-}$ $P < 2.2e-16$, *Hormad1*$^{-/-}$ vs. *Ankrd31*$^{-/-}$ *Hormad1*$^{-/-}$ $P = 2.49e-11$, early pachytene, wt vs. *Ankrd31*$^{-/-}$ $P = 0.0001234$, *Ankrd31*$^{-/-}$ vs. *Iho1*$^{C7\Delta/C7\Delta}$ $P = 1.1e-10$. **C** Immunostaining in leptotene spermatocytes from adult mice. Bars, 10 µm. **E** Radiograph of immunoprecipitated and radio-actively labeled SPO11-oligo complexes from testes of adult mice. Bar, SPO11-specific signal, asterisk, nonspecific labelling, and arrowhead, immunoglobulin heavy-chain. Radioactive signals were background-corrected (*Iho1*$^{-/-}$, signal=0) and normalized to wild-type control (1). Means and standard deviations are from $n = 2$ biological replicates. **F** Schematic summary of phenotypes caused by the disruption of IHO1-HORMAD1 complex and/or ANKRD31. **G** Model for the assembly of DSB-factor clusters on axis. Black arrows represent promotion of (i) IHO1 phosphorylation and (ii) seeding or (iii) growth of DSB-factor clusters by CDC7-DBF4, IHO1 (in particular, phosphorylated IHO1 C-terminus) and ANKRD31, respectively. See also related Supplementary Fig. 10 and 11 and Supplementary Table 6. Source data are provided as a Source Data file.

and 12.5%, n = 592 clusters in *Hormad1*$^{-/-}$ *Ankrd31*$^{-/-}$), which may indicate spontaneous aggregation of DSB factors if both IHO1-HORMAD1 and ANKRD31 functions are lost.

In addition, DSB activity was more substantially disabled by coincident disruption of ANKRD31 and the IHO1-HORMAD1 complex. Chromatin-associated levels of ɣH2AX were strongly reduced in *Iho1*$^{C7\Delta/C7\Delta}$ *Ankrd31*$^{-/-}$ and *Hormad1*$^{-/-}$ *Ankrd31*$^{-/-}$ as compared to *Iho1*$^{C7\Delta/C7\Delta}$, *Ankrd31*$^{-/-}$ and *Hormad1*$^{-/-}$ single mutant spermatocytes in leptotene to early zygotene stages (Supplementary Fig. 10C, D), where ɣH2AX levels principally reflect ATM signaling from DSBs[71]. *Iho1*$^{C7\Delta/C7\Delta}$ *Ankrd31*$^{-/-}$ and *Hormad1*$^{-/-}$ *Ankrd31*$^{-/-}$ spermatocytes did not progress beyond a defective zygo-pachytene-like stage where fully formed axes remained largely unsynapsed, yet axes often engaged in multiple apparently nonhomologous entanglements (Supplementary Fig. 10E). In significant fractions of these spermatocytes, ɣH2AX accumulated in one or a few large chromatin domains (Supplementary Fig. 10E, F).

These phenotypes closely resembled the phenotypes of *Spo11*$^{-/-}$ and *Iho1*$^{-/-}$ spermatocytes, where lack of programmed DSBs prevented homology search, and where large ɣH2AX-rich chromatin domains, called pseudo-sex bodies, frequently formed due to ATR signaling from unsynapsed axes and/or sporadic SPO11-independent DSBs[71,72]. Consistent with a reported HORMAD1 role in pseudo-sex body formation in DSB deficient meiocytes[22,32], ɣH2AX-rich chromatin domains were less frequent in *Hormad1*$^{-/-}$ *Ankrd31*$^{-/-}$ than *Iho1*$^{C7\Delta/C7\Delta}$ *Ankrd31*$^{-/-}$, *Spo11*$^{-/-}$ or *Iho1*$^{-/-}$ spermatocytes.

Their apparent similarities to *Spo11*$^{-/-}$ and *Iho1*$^{-/-}$ spermatocytes raised the possibility that *Hormad1*$^{-/-}$ *Ankrd31*$^{-/-}$ and *Iho1*$^{C7\Delta/C7\Delta}$ *Ankrd31*$^{-/-}$ spermatocytes lacked programmed DSBs. Indeed, DSB foci, as detected by RAD51, DMC1 and RPA2 staining, were rarely, if at all, present in these double mutants (Fig. 7D and Supplementary Fig. 11). Further, SPO11-oligo complexes[58] were not detected in testes of *Iho1*$^{C7\Delta/C7\Delta}$ *Ankrd31*$^{-/-}$ and *Hormad1*$^{-/-}$ *Ankrd31*$^{-/-}$ mice, confirming a lack of DSB activity (Fig. 7E). Therefore, we conclude that ANKRD31 and the IHO1-HORMAD1 complex are redundantly required for meiotic DSB formation and recombination initiation.

## Discussion

Orderly synapsis of homologous chromosomes is enabled by DSB formation on unsynapsed chromosome axes in mammals, yet the mechanism that ensures this appropriate spatial organization of DSB activity is poorly explained. However, recent insights into the mechanism of DNA-bound clustering of DSB factors, together with our data, allows us to propose a model for the axial assembly of the DSB-machinery (Fig. 7F, G).

### Enabling seeding and growth of DSB-factor clusters on chromosome axes

Multivalent protein-protein and protein-DNA interactions cooperatively condense the budding yeast orthologs of MEI4 (Mei4), REC114 (Rec114) and IHO1 (Mer2) into DNA-bound co-clusters in vitro[28]. Mer2 also efficiently binds nucleosomes in vitro[29,73]. DNA binding of DSB factors also shows considerable conservation[73,74]. Thus, the in vitro clusters of DNA-bound DSB-factors of budding yeast likely model in vivo DSB-factor clusters which enable DSB activity in diverse taxa including mammals. However, assembly on DNA cannot fully explain the situation in vivo: DSB-factor clusters preferentially assemble on chromosome axes rather than on bulk chromatin[13,20,22,36,43,57].

Our analysis suggests that ANKRD31 and IHO1-HORMAD1 interaction play crucial and functionally synergistic roles in the axial assembly of the DSB-machinery. Whereas ANKRD31 is not an essential component of DSB-factor clusters[23,24], it directly interacts with at least three essential DSB factors, REC114[23,24], IHO1[24] and MEI1 (this study). Hence, ANKRD31 may enhance inter-molecular connections between DSB factors during clustering. Accordingly, ANKRD31 supports both the establishment and subsequent growth of DSB-factor clusters.

In contrast, the IHO1-HORMAD1 interaction seems to specifically enable the axial seeding, but not the growth of DSB-factor clusters. The underlying reason could be that IHO1-HORMAD1 interaction axially anchor DSB-factor clusters without influencing their internal architecture. IHO1 recruitment to chromosome axes relies on a stable IHO1-HORMAD1 interaction that does not require other DSB factors. Furthermore, IHO1 requires its conserved C-terminal acidic patch for interaction with HORMAD1, but IHO1 requires neither its C-terminus nor HORMAD1 for interactions with itself, REC114, MEI1 and ANKRD31 (this study)[22,24,75] indicating distinct molecular requirements for IHO1 interaction with HORMAD1 and DSB-factors. Hence, HORMAD1-mediated assembly of an axial IHO1 platform could enable efficient seeding of cytological DSB-factor clusters by providing arrays of anchor sites for REC114, MEI1 and ANKRD31 recruitment.

There is an ongoing discussion to what extent multivalent low affinity/low specificity interactions versus high specificity interactions contribute to biogenesis of membraneless subcellular compartments[76]. The paradigm of HORMAD1-IHO1 interaction-promoted seeding of DSB-factor clustering suggests that both high specificity interactions (IHO1-HORMAD1) and multivalent low specificity interactions (DNA-driven condensation) can significantly contribute to mesoscale macromolecular assemblies representing subcellular compartments.

IHO1 could support axial seeding of DSB-factor clusters by distinct mechanisms. It is likely that the dynamics of DSB-factor cluster formation are influenced by a balance between cluster nucleation,

dissolution, fusion and growth by incorporation of components from soluble pools whose limited size results in a competition between clusters[28]. Axial IHO1 may directly initiate biochemical nucleation of DSB-factor clusters. A nonexclusive alternative is that DSB-factor clustering is nucleated across chromatin, but that the capture of resultant proto-clusters by axial IHO1 platforms shifts the balance of post nucleation cluster dynamics, thereby stabilizing clusters and/or enabling efficient establishment of cytologically discernible clusters − which likely represent functionally competent DSB machineries. Further, axial IHO1 platforms may attenuate the effect that DSB-triggered negative feedback has on DSB-factor clusters by supporting both existing clusters and/or de novo cluster assembly from soluble pools of DSB factors.

The IHO1 C-terminus is not only required for the efficient formation of DSB-factor clusters but also DSBs, indicating that the IHO1-HORMAD1 interaction is a key guarantor of DSBs on unsynapsed axes. Nevertheless, if the axial IHO1 anchor is disrupted by deletion of IHO1 C-terminus or HORMAD1, functional DSB-factor clusters and dependent DSBs still form on axes, albeit at low frequency. One or more DSB factors and axis components may engage in cooperative low affinity interactions, which permits seeding of cytological DSB-factor clusters with low efficiency. Due to their low numbers, successfully seeded clusters may grow bigger than normal by sequestering soluble DSB factors. These DSB-factor clusters are strictly dependent on ANKRD31, which may reflect a need for high valency of interactions between DSB factors to allow stabilization of their heteromeric clusters in the absence of an axial IHO1 platform.

LaDSB-factor clusters on PARs and PAR-like regions do not require HORMAD1 or IHO1, but critically depend on ANKRD31, REC114 and MEI4[21,23–25]. Thus, instead of a stable IHO1-HORMAD1 interaction, cooperative protein-protein and protein-DNA interactions may take the lead in seeding laDSB-factor clusters, resembling in vitro condensate formation. PARs and PAR-like regions are rich in mo-2 minisatellite repeats[25], which may provide arrays of binding sites for efficient seeding of DSB-factor clusters independent of IHO1-HORMAD1 interaction.

**Phosphorylation of IHO1 C-terminus may contribute to the control of DSB activity**

The IHO1-HORMAD1 interaction critically depends on serines in positions 569 and 570 in the IHO1 C-terminus. Phosphorylation of these serines was detected in vivo[42], but phosphorylation of IHO1 is not essential for IHO1-HORMAD1 interaction in vitro, questioning the role of phosphorylation in vivo. Importantly, whereas in vivo phosphorylation of IHO1 C-terminus occurs in the absence of axial recruitment of IHO1 in *Hormad1*[−/−], IHO1 phosphorylation correlates with IHO1 accumulation on chromosome axes in wild type. It follows that IHO1 phosphorylation may enhance IHO1-HORMAD1 interaction. Phosphorylation may enable efficient binding of the IHO1 C-terminus to HORMAD1 and/or it may enable conformation changes that enhance HORMAD1-binding affinity of IHO1 regions away from the C-terminus. Consistent with these hypotheses, we found that both IHO1 phosphorylation and axial recruitment are enhanced by DDK activity in mice. DDK seems to promote recombination initiation, not only independently, but also by contributing to the control of IHO1 axial loading. In contrast to the role of DDK in mice, DDK-mediated phosphorylation of the budding yeast Mer2 (IHO1 ortholog) is required for DSB activity but not axial recruitment of Mer2[20,49–51]. Thus, DDK seems to be important for recombination initiation in both mice and yeast, but there is an apparent divergence in Mer2/IHO1 functions that are controlled by DDK. We urge caution in these conclusions because our data do not directly show that DDK-dependent phosphorylation of IHO1 C-terminus promote IHO1-HORMAD1 interaction. Furthermore, the phenotypes of phosphomimetic IHO1 mutations raise the possibility that negative charges in positions 569 and 570 in the IHO1

C-terminus hinder IHO1-HORMAD1 interaction and IHO1 axial recruitment. Therefore, further investigations will be necessary to understand the consequences of IHO1 phosphorylation, and the role of DDK in it.

**IHO1-HORMAD1 interaction supports error correction of synapsis**

Delays in homolog synapsis − which occur in several mouse models, i.e. *Ankrd31*[−/−], *Tg(Spo11β)*[+/− 39] or *Spo11*[+/−] − are thought to be remedied by persistent DSB activity on asynaptic axes[37–40]. Curiously, *Iho1*[C7Δ/C7Δ] meiocytes show enduring asynapsis without associated accumulation of DSBs, indicating that maintenance of DSB activity in asynaptic regions depends on IHO1-HORMAD1 interaction. The IHO1-HORMAD1 interaction seems to enhance not only the formation of DSB-factor clusters, but also their resistance to destabilizing negative feedback from previously formed DSBs after leptotene. We hypothesize that loss of the latter function considerably contributes to synapsis defects in *Iho1*[C7Δ/C7Δ] meiocytes by preventing asynapsis-enabled maintenance of DSB activity and downstream correction of synapsis errors.

**Evolutionary comparison suggests that axial localization of DSB-machinery enables coordination of DSB activity and homolog pairing**

There is considerable evolutionary divergence in the sequences and functions of DSB factors and the spatiotemporal control of DSBs. Co-evolution of distinct features of DSB-machinery and DSB regulation may indicate their functional coupling, hence inter-taxa comparisons reveal general principles of DSB control.

Interactions between HORMA domain proteins and Mer2/IHO1-family proteins are thought to provide axial anchors for the DSB-machinery in multiple species[20,22,29–31,36,43]. Yet, redundancies in mechanisms that link DSB-machinery to axes diminish the importance of the interaction between Mer2/IHO1- and HORMAD1-family proteins in some taxa[30,36]. Consistent with this principle, a HORMAD1 ortholog has not been reported in the fungus *Sordaria macrospora*, and an acidic patch is absent from the C-terminus of the *Sm*Mer2[14]. Furthermore, *Sm*Mer2 exhibits unique behavior among Mer2/IHO1-family proteins by being present on chromosomes from early meiosis till early post-meiotic divisions and by having post-DSB roles in homolog pairing, recombination and late meiotic chromosome compaction in addition to being essential for DSBs. In contrast, Mer2/IHO1-family proteins of all other examined taxa diminish from chromosomes following synapsis and likely lack major functions besides DSB formation[18,22,31,34,36,43]. Thus, while *Sm*Mer2 retained its axis−associated role in enabling DSB formation, it seemingly acquired new post-DSB roles that co-evolved with (i) an altered control of its chromatin binding, (ii) the loss of its C-terminal acidic patch, and (iii) a presumed lack of a HORMAD1 ortholog.

A unique paradigm is presented by the nematode *Caenorhabditis elegans* (worm hereafter), which lacks a recognizable Mer2/IHO1 ortholog but possesses REC114 (DSB-1 and 2) and MEI4 (DSB-3) orthologs[14,15], which are crucial for DSBs[15,77,78]. Unlike DSB factors of most other taxa, worm DSB-1/2/3 form clusters on bulk chromatin instead of axes[15,77,78]. The absence of a Mer2/IHO1-family protein may deprive DSB-factor clusters from an axial anchor, thereby preventing their axial enrichment. Judging from the localization of DSB-factors, it is conceivable that DSBs form off-axis in worms[15]. Yet, recombination foci seemingly associate with the axis[79], suggestive of a post-DSB mechanism that recruits recombination intermediates to axes[80]. Consistently, irradiation-induced DSBs − which are not confined to axes − give rise to axis-associated recombination foci in mice, and a similar configuration may be present in irradiated worms[34,63,81]. The existence of post-DSB mechanisms for axial recruitment of recombination foci raises the question why DSB activity is focused on axes in most taxa.

In most taxa, recombination initiation by DSBs is required for synapsis of homolog axes. Therefore, generation of DSBs − which are potentially genotoxic − is not beneficial once synapsis is achieved. Axial association places DSB-factors in the physical context of the SC, allowing DSB activity to be controlled by synapsis. DSB formation is maintained in unsynapsed regions, where DSBs are needed for promoting synapsis, whereas synapsis depletes DSB-factor clusters and ends DSB activity[34,35,37−39]. Unlike most studied taxa, worms synapse homolog chromosomes using pairing centers rather than DSBs[82−84]. Nevertheless, worms still require DSBs for CO formation[84]. Due to altered requirements, worms maintain competence for DSB formation on synapsed chromosomes, and they link depletion of DSB-factor clusters and cessation of DSB activity to the formation of COs instead of synapsis[77,78].

We speculate that axial accumulation of DSB factors is not conserved in worms because unique alterations in homolog pairing mechanisms make it unnecessary to coordinate DSB activity with synapsis in this taxon. Accordingly, one reason why organisms that rely on DSBs for synapsis may accumulate DSB-factors on the axes − besides priming axial localization of recombination intermediates − is that axis association of DSBs enables appropriate coordination of DSB activity with homolog synapsis, which both aids efficient homolog pairing and prevents excessive DSBs during meiosis.

## Methods

### Animal experiments

Gonads were collected from mice after cervical dislocation. Most cytological experiments of spermatocytes were carried out on samples collected from adult mice unless indicated otherwise. We used testes of juvenile mice (12-13 days old) for most biochemical experiments to enrich pre-pachytene spermatocytes and to ensure that the cellular compositions of testes were similar in wild-type and meiotic recombination mutant mice. The mid pachytene stage, where most recombination defects trigger apoptosis[52], is reached by a majority of spermatocytes at around 14 days of age during the first developmental wave of meiosis in mice. Therefore, testes are directly comparable in wild-type and meiotic mutant mice at 12-13 days of age.

The mice were kept in the barrier facility in individually ventilated cages at 22−24 °C and 50−55% air humidity with 14-h light/10-h dark cycle. The feed was a rat−mouse standard diet in the form of pellets. The stocking density in the used cage type IIL was maximum five mice. Hygiene monitoring was carried out according to FELASA guidelines. All procedures pertaining to animal experiments were approved by the Governmental IACUC ("Landesdirektion Sachsen") and overseen by the animal ethics committee of the Technische Universität Dresden. The license numbers concerned with the present experiments with mice are TVV 2014/17, TV A 8/2017, TV A 23/2017, and TV vG 3/2022.

In addition to newly generated $Iho1^{C7\Delta/C7\Delta}$ mice we used previously published $Ankrd31^{-/-24}$, $Iho1^{-/-22}$, $Hormad1^{-/-32}$, $Dmc1^{-/-85}$ and $Spo11^{-/-7}$ mice.

### Generation of $Iho1^{C7\Delta}$ mice

$Iho1^{C7\Delta}$ mutant line was generated using CRIPSR/Cas9 genome editing[86], targeting exon7 of $Iho1$ gene. A mixture of gRNA: GGATTTTGATAGCAGCGATGATA (12.5 ng/ml, IDT), designed using the online platform at http://crispr.mit.edu/guides and https://gt-scan.csiro.au/gt-scan[87], and Cas9 nuclease mRNA (50 ng/ml), prepared as described before[24], was injected into pronucleus/cytoplasm of fertilized oocytes. The oocytes were subsequently transferred into pseudopregnant recipients. A founder mouse that was heterozygote for an insertion of T nucleoside that caused a premature stop codon was bred with C57BL/6JCrl wild-type mice to establish mouse lines. All experiments reported in the manuscript are based on samples from mice that were derivative of this founder mouse after at least three backcrosses.

**Cas9 mRNA production**. To prepare Cas9 mRNA, we first used the restriction enzyme PmeI to linearize the plasmid MLM3613 (Addgene #42251)[86] that harbors a codon optimized Cas9 coding sequence and a T7 promoter for Cas9 mRNA in vitro synthesis. We then used the linearized MLM3613 as template to synthesize the 5′ capped and 3′ polyA-tailed Cas9 mRNA using the mMESSAGE mMACHINE® T7 Ultra Kit (ThermoFisher, cat no: AM1345) according to the manufacturer's instructions.

### Genotyping

Tail biopsies were used to generate genomic DNA by overnight protease K digestion at 55 °C in lysis buffer (200 mM NaCl, 100 mM Tris-HCl pH 8, 5 mM EDTA, 0.1% SDS), followed by heat inactivation for 10 min at 95 C. Genotypes were identified either by heteroduplex mobility assay (HMA) or by the use of mismatch primers in genotyping PCRs. For HMA, PCRs were carried out with CATGACCACCA-GAAGCGTCA and AATGTTTTCACCAAGGACATAC primers, which annealed to sequences flanking the mutated site in the Iho1 gene. Genotyping PCR products or one-to-one mixtures of genotyping PCRs and PCRs of wild type genomes were supplemented with EDTA in 10 mM final concentration. After a 2-minute denaturation cycle at 98 °C and a 30-minute annealing cycle at 25 °C, electrophoresis was performed in 8% polyacrylamide gels that were prepared with TBE buffer. As an alternative strategy to HMA, and to allow analysis of genotyping PCRs on agarose gels we also carried out PCRs with primers that annealed to the site of mutation in the $Iho1$ gene. Due to only a single base difference in the sequence of wild-type and mutant loci, primers that annealed to the site where the mutation was introduced enabled amplification from both wild-type and mutant loci. To better distinguish between wild-type and mutant templates we introduced mismatches into both wild-type and mutant specific primers. CAGA-GATCAAAGAGAGGTGG and CATCGCTGCTATCAA<u>G</u>ATC (mismatch nucleotide underlined) primers allowed high specificity amplification of a 385 bp product from the wild type allele. CAGAGATCAAAGA-GAGGTGG and CATCGCTGCTATCAAA<u>A</u>GTC (mismatch nucleotide underlined) allowed high specificity amplification of a 386 bp product from the $Iho1^{C7\Delta}$ allele. Geneious version 5.6.3 was used to design primers for genotyping.

### Testis electroporation

To overexpress tagged wild type and mutant proteins in spermatocytes, we injected 6−8 µl of an expression vector (3-5 µg/µl) under the control of a CMV promoter into the rete testis of live juvenile mice (13 days postpartum/dpp) according to published protocol[88,89]. 45 minutes after injection, testes were held between tweezer type of electrodes (CUY650P5, Nepagene) and in vivo electroporation was carried out with four 50-ms pulses (35 Volts) with 950 ms intervals, then four equivalent pulses with opposite polarity (NEPA21 Electroporator, Nepagene).

### Protein extracts, immunoprecipitation and western blotting

**Total protein extraction**. Testes of 8dpp − in the case of testis cell cultures − or 12-13 dpp juvenile mice were collected, and tunica albuginea was removed. Extracts were prepared from testes either after in vitro culture (two days of culture with or without CDC7 inhibitors, see details later), or testes were immediately used, or they were flash frozen in liquid nitrogen for later use. Frozen testes were thawed on ice for 10-15 minutes before use. Fresh or thawed tissue were homogenized with the help of disposable tissue grinder pestle (VWR, BELAF199230000) in resuspension buffer (50 mM Tris pH=7.4 150 mM NaCl, supplemented with protease and phosphatase inhibitors: 1 mM Phenylmethylsulfonyl Fluoride (PMSF), complete EDTA-free Protease Inhibitor Cocktail tablets (Roche, 11873580001), 0.5 mM Sodium orthovanadate, Phosphatase inhibitor cocktail 1 (Sigma, P2850) and Phosphatase inhibitor cocktail 2 (Sigma, P5726), protease and

phosphatase inhibitor cocktails were used at concentrations recommended by the manufacturers). Testes homogenates were mixed 1:1 ratio with 2X lysis buffer (Tris 50 mM pH=7.4, 850 mM NaCl, 2% Triton X-100, 2% NP40, 1% Sodium deoxycholate/NaDOC, 20 mM MgCl2, supplemented with protease and phosphatase inhibitors as above). Testis homogenates were lysed with the help of overhead rotator for 60 min at 4 C in the presence of benzonase (Merck Millipore) to digest DNA during lysis. Total testis lysates were mixed with 2x Laemmli sample buffer (with 10% b-Mercaptoethanol) 1:1 ratio and incubated for 10 min at 95 C. Samples were run on 10% gel with using 10 × 10.5 cm gel cassette (Cytiva miniVE Vertical Electrophoresis System) to improve separation of phosphorylated and non-phosphorylated IHO1 bands.

**Fractionation of testis extracts based on Triton X-100 solubility.** Testes were homogenized with the help of disposable tissue grinder pestle in resuspension buffer (as described for total extracts). Big tissue chunks were additionally disrupted by 200 μL pipette tip with a cut end. Homogenized testes were divided into 2 parts – ¼ of the homogenate for total extract and ¾ for fractionation. Both aliquots were mixed 1:1 with 2x Triton X-100 buffer (Tris 50 mM pH=7.4, 150 mM NaCl, 0.6 % Triton X-100, supplemented with protease and phosphatase inhibitors) and incubated at 4 °C for 30 min, constantly mixed with overhead rotator. Thereafter, lysed homogenates intended for total extracts were placed on ice until further processing, and aliquots intended for fractionation were centrifuged at 4 °C for 10 min at 16000 g. Supernatants were collected to a new tube as soluble fraction. Pellets containing insoluble fractions were resuspended in the same buffer in the same volume as the supernatant. The unfractionated extracts and both the Triton X-100 soluble and insoluble fractions were mixed with 2x Lysis buffer (850 mM NaCl, 50 mM Tris-HCl pH 7.5, 2% Triton-X100, 2% NP-40, 1% NaDOC, 1 mM MgCl2 supplemented with protease inhibitors and benzonase). After 1 h of incubation, unfractionated extracts and soluble and insoluble fractions were mixed with 2x Laemmli sample buffer (with 10% b-Mercaptoethanol) 1:1 ratio and incubated for 7 min at 95 °C.

**IHO1 dephosphorylation assay.** Total testis lysates were prepared as described above but phosphatase inhibitors were omitted. Each pair of testes from one juvenile male mouse was lysed in 50 μL 1x lysis buffer. After 1.5 h of incubation sample was centrifuged 16 000 g for 10 min. Then supernatant was diluted 2 times with phosphatase dilution buffer: 100 mM NaCl, 50 mM Tris-HCl pH 7.5 supplemented with protease inhibitors. Afterwards, lysate was split into 4 aliquots (35 μL each). All aliquots were supplemented with 10X NEBuffer for Protein Metallo-Phosphatases (PMP) and 10X MnCl2 (NEB). The first aliquot (Untreated control) was supplemented with phosphatase inhibitors (PPase inhibitors): 0.5 mM sodium orthovanadate, phosphatase inhibitor cocktail 1 (Sigma, P2850) and phosphatase inhibitor cocktail 2 (Sigma, P5726). They were used at concentrations recommended by the manufacturers. After that 2x Laemmli buffer (with 10% b-Mercaptoethanol) was added and sample was boiled 95 °C for 7 min. The second (PPase inhibitors only) and third (PPase inhibitors + Lambda PPase) aliquots were also supplemented with PPase inhibitors. 2 μL of Lambda Protein Phosphatase (NEB, P0753S) was added to aliquots 3 and 4 (PPase only). Final volume of each sample was 50 μL. Aliquots 2-4 were incubated at 30 °C for 1.5 h. After incubation, 2xLaemmli buffer (with 10% b-Mercaptoethanol) was added 1:1 ratio and samples were incubated at 95 °C for 7 min.

**IHO1 phos-tag gel preparation and Western Blot.** For analysis on phos-tag gel, total lysates of testis were prepared as described above. For preparation of phos-tag gels the following reagents were used (indicated final concentrations): 8% acrylamide/bis-solution (ROTH), 20 μM phos-tag acrylamide AAL-107 aqueous solution (Fujifilm),

40 mM MnCl2 (Sigma), 0.04% SDS (ROTH), 0.04% APS and TEMED (Bio-Rad)(ROTH), 375 mM Tris-HCl pH 8.8 (ROTH). Following electrophoresis, gels were incubated 3×10 minutes in Tris/Glycine transfer buffer without methanol and supplemented with 10 mM EDTA. After treatment, proteins were transferred on PVDF-membrane with Tris/Glycine transfer buffer supplemented with 20% Methanol for 2.5 h (300 mA). PVDF-membranes were blocked for 1 h with 5% non-fat dry milk in TBS-T, followed by incubation overnight with primary antibodies.

**IHO1 and HORMAD1 immunoprecipitation.** Total testis lysates were spun down for 10 min at 16.000 g after lysis. Supernatants were collected to a new tube and diluted two times with 1x IP lysis buffer (50 mM Tris-HCl pH 7.4, 500 mM NaCl, 1% Triton X-100, 1% NP40, 0.5% NaDOC, supplemented with protease and phosphatase inhibitors). To reduce non-specific immunoprecipitation we precleared lysates by adding 1 mg Dynabeads Protein A (Invitrogen), and mixing them for 4 hours at 4 °C in an overhead rotator. After pre-clearing, protein extracts were incubated with affinity beads at 4 °C overnight. Either rabbit anti-IHO1 (3 μg) antibodies or rabbit anti-HORMAD1 (3 μg) antibodies were cross-linked to 1.5 mg of Dynabeads Protein A (Invitrogen) using 20 mM Dimethyl Suberimidate according to standard protocols to prepare affinity beads. Following overnight incubation in the protein extracts, affinity beads were washed three times with 1x IP lysis buffer. Immunoprecipitated materials were eluted by incubating the beads in 50ul 1x Laemmli sample (supplemented with 5% b-Mercaptoethanol) buffer for 10 min at room temperature, agitated 2-3 times with vortex in between. The resulting eluates were analyzed with SDS-PAGE and immunoblotting using standard methods. Briefly, proteins from extracts were separated on SDS polyacrylamide gels and blotted onto Nitrocellulose membrane (Hybond ECL Cat#RPN2020D, GE Healthcare). Membranes were blocked using Skimmed Milk 5%, 0.05% Tween, TBS blocking solution. Band intensities were measured using ImageJ.

**Expression and purification.** Codon-optimized synthetic dsDNA (Geneblocks) for IHO1 and HORMAD1 were synthesized by IDT (Leuven, Belgium). Constructs were cloned into pCOLI or pLIB vectors for E.coli or insect cell expression, respectively[90,91].

For insect cell expression, virus was generated from Sf9 cells using standard protocols. All the IHO1 truncations were expressed with the 3 C HRV cleavable N-terminal MBP tag in suspension culture of High Five™ Cells (Invitrogen) in Sf-900™ III SFM medium, infected with virus (diluted 1:000), for 48 h at 27 °C. Cell pellets were washed with 1× PBS and resuspended in lysis buffer (50 mM HEPES pH 7.5, 300 mM NaCl, 5% glycerol, 0.1% Triton-X 100, 1 mM MgCl₂, 5 mM β-mercaptoethanol, AEBSF 25 μg/mL). For cell lysis, sonication at 30% of power was used, and lysate was cleared by centrifugation at 40000 g for 40 min. Cleared lysate was incubated with benzonase (Sigma-Aldrich) at 8 °C for 20 min and then applied on a 5 ml MBP-trap column (Cytiva), wash with lysis buffer and eluted with 1 mM maltose solution based on lysis buffer. The elution fractions with highest content of IHO1 truncations were applied to Resource Q column (GE healthcare) equilibrated with (50 mM HEPES pH 7.5, 100 mM NaCl, 5% glycerol, 1 mM TCEP). The proteins were eluted with the gradient of NaCl up to 1 M. Proteins after ion-exchange chromatography were concentrated on the Pierce™ Protein Concentrator PES (30 K MWCO, 5-20 ml) and applied to the Superdex200 10/300 Increase column (Cytiva) equilibrated with 50 mM HEPES pH 7.5, 300 mM NaCl, 5% glycerol, 1 mM TCEP.

SUMO-His N-terminally tagged HORMAD1 HORMA (amino acids 1-235) was expressed in chemically competent E.coli BL21(DE3) cells overnight at 18 °C with induction of protein expression by 250 μM IPTG. The subsequent purification of the protein was carried out according to the same protocol as with IHO1 truncations, except for the affinity chromatography step. For this cleared lysate of the cells was applied to

the Talon column (Cytiva), equilibrated with the lysis buffer (50 mM HEPES pH 7.5, 300 mM NaCl, 10% glycerol, 1 mM MgCl$_2$, 0.1% Tween20, 5 mM imidazole, 5 mM β-mercaptoethanol), wash with two steps of lysis buffer with higher concentrations of imidazole (10 and 20 mM) and eluted with lysis buffer containing 250 mM imidazole.

**Amylose pulldown assay.** Amylose pulldown assay was performed with Amylose Sepharose beads (New England BioLabs) in a pulldown buffer (20 mM HEPES pH 7.5, 100 mM NaCl, 5% glycerol, 1 mM TCEP, 0.1% Tween20). The beads were pre-blocked with 1 mg/ml BSA in pulldown buffer for 2 h at 8 °C washed twice with 500 µL and then resuspended in equal volume of pulldown buffer. The MBP-tagged IHO1 truncations (residues 359-574, 440-574, 358-567, 439-56) were incubated at 1 µM concentration as baits with 12 µM of SUMO-His tagged HORMAD1 HORMA (residues 1-235) for 30 min on ice. Input samples were mixed with 6x SDS-PAGE sample buffer and loaded on the 10% SDS-PAAG (5 µL for anti-MBP and anti-HORMAD immunostaining and 10 µL for Coomassie staining). Resuspended beads were incubated with protein mix on shaker at 8 °C for 30 min and then washed twice with 100 µL of the pulldown buffer. Elution was performed by adding 40 µL of 1x SDS-PAGE sample buffer (10 µL was loaded for each immunostaining and 20 µL for Coomassie staining). After SDS-PAGE, one gel was stained with Coomassie Blue (Blaue Jonas Stain, GRP GmbH) and proteins from another gel were transferred onto nitrocellulose membrane. After Western blotting the membrane was blocked in 5% nonfat milk in PBS with 0.1% Tween-20 (PBST) and incubated with primary antibodies overnight at 4 °C. Rabbit polyclonal anti-HORMAD1 antibodies (Abcam) were used at the concentration 1 µg/µl. After incubation with primary antibodies, the membrane was washed 3 times with excessive amount of PBST, incubated with secondary anti-rabbit IgG antibodies and washed with PBST. Detection of the proteins was performed using ECL Prime Western Blotting Detection Reagent.

## Analysis of DDK-mediated phosphorylation of IHO1 C-terminal peptide

**In Vitro phosphorylation of IHO1 peptides by DDK.** We prepared 100-µl reaction mixes that contained 1x kinase reaction buffer (5x buffer supplied by Promega, V5088), 2 mM DTT, 100 µM Ultrapure ATP, 0.7 µg of recombinant full-length human CDC7/DBF4 kinase (Promega, V5088) and C-terminal peptides of IHO1 (NLLCDPDFDSSDDNF ending with either NH2 or COOH groups) at 0.4 mM final concentration. Reactions were incubated at 30 °C for 120 min and terminated by freezing on dry ice before mass spectrometry.

**Phospho peptide preparation and SEC separation for mass spectrometry:.** Frozen samples were thawed and 29 µL ACN with 0.43% TFA was added to 100 µL of sample to achieve a concentration of 22.5% ACN and 0.1% TFA in total. The samples were left on room temperature for 5 min and centrifuged at 14 000 g for 10 min to remove possible precipitation products. 16 µL of the sample was in injected into a Dionex Ultimate 3000 (Thermo Fisher Scientific, Vienna, Austria) using a Cytiva Superdex 30 Increase Precision 3.2/300 GL Prepacked SEC Column (Cytiva, Marlborough, MA, USA), 77.5% H$_2$O, 22.5%, 0.1% TFA as mobile phase and was separated with a flow rate of 25 µL min$^{-1}$. Absorption of UV light at 215 nm detected a major peak at 56.9 min corresponding to a molecular mass of about 1500-2000 Da. The collected fraction at 54-60 min was dried and used for HPLC-MS analysis.

**HPLC-MS analysis:.** For the identification of the peptides a Dionex Ultimate 3000 RSLCnano System coupled to an Orbitrap Eclipse Tribrid Mass Spectrometer (both Thermo Fisher Scientific, Vienna, Austria) was used. The dried samples were dissolved in 25 µL mobile phase A (98% H$_2$O, 2% ACN, 0.1% FA), shaken for 10 min at 30 °C and centrifuged at 14000 g for 1 min. 2 µL were injected onto a PepMap RSLC

EASY-Spray column (C18, 2 µm, 100 Å, 75 µm x 15 cm, Thermo Fisher Scientific). Separation occurred at 300 nL min$^{-1}$ with a flow gradient from 2-35% mobile Phase B (2% H$_2$O, 98% ACN, 0.1% FA) within 25 min resulting in a total method time of 55 min. Mass spectrometer was operated with the FAIMS Pro System in positive ionization mode at alternating CV −60 and −75. The scan range was 350-2000 m z$^{-1}$ using a resolution of 60,000 @200 m z$^{-1}$ on MS1 level. Isolated peptides were fragmented using CID at a collision energy of 30% and fragments were analyzed in the Orbitrap with a resolution of 30,000 @200 m z$^{-1}$.

**Peptide identification:.** Peptides were identified using Proteome Discoverer 2.5 (Thermo Fisher Scientific) employing the Sequest HT search engine[92]. Phosphorylation sites were determined using the implemented ptmRS node. Raw files were searched against the mouse proteome derived from UniProt (*Mus musculus*, 17082 reviewed entries, canonical and isoforms) and the additional C-terminal Iho1 peptide NLLCDPDFDSSDDNF. Phosphorylation (STY) and Amidation (C-terminal) were set as variable modifications.

**Spo11-oligo.** To measure levels of SPO11-oligonucleotide complexes in testes, we carried out SPO11 immunoprecipitations and SPO11-oligonucleotide detection as published previously[32,58]. Both testes from one adult mouse were used for each experiment.

**Yeast two-hybrid (Y2H) assay.** Yeast two-hybrid experiments with mouse proteins were performed as described previously with minor modifications[22]. Pairwise interactions were carried out in the Y2HGold Yeast strain (Cat. no. 630498, Clontech). For transformation of Y2HGold with bait and prey vectors, yeast were grown in 2xYPDA medium overnight at 30 °C, 200 rpm shaking. Afterward, yeast cells were diluted to 0.4 optical density (OD, measured at 600 nm) and incubated in 2x YPDA for 5 h at 30 C, 200 rpm shaking. Cells were harvested, washed with water and re-suspended in 2 mL of 100 mM lithium acetate (LiAc). 50 µL of this cell suspension was used for each transformation. Transformation mix included 1 µg of each vector (bait and prey), 60 µL of polyethylene glycol 50% (w/v in water), 9 µL of 1.0 M LiAc, 12.5 µL of boiled single-strand DNA from salmon sperm (AM9680, Ambion), and water up to 90 µL in total. The transformation mix was incubated at 30 °C for 30 min, and then at 42 C for 30 min for the heat shock. The transformation mix was removed following centrifugation at 1000 g for 10 min, and then cells were resuspended in water, and plated first on -Leu -Trp plates to allow selective growth of transformants. After 2-3 days of growth, transformants were plated both on -Leu -Trp and -Leu -Trp -Ade -His plates for 2-7 days to test for interactions. We followed the manufacturer's instructions for media and plate preparation. Yeast two-hybrid essays of Arabidopsis proteins were performed as described previously[93]. Specifically, the relevant Gal4-AD and BD constructs were co-transformed into the auxotrophic yeast strain AH109 using the polyethylene glycol/lithium acetate method according to the manufacturer's manual (Clontech). Yeast cells haboring the relevant combinations of constructs were dotted on plates with double (-Leu-Trp), triple (-Leu-Trp-His) and quadruple (-Leu-Trp-His-Ade) synthetic dropout medium to assay growth.

## Immunofluorescence microscopy

**Preparation of spermatocyte spreads.** Preparation of nuclear surface spreads of spermatocytes was carried out according to earlier described protocols with minor modifications[24,94]. Testes were minced with scalpels in PBS pH 7.4 and collected into a clean tube. The testis suspension was left standing for few minutes to allow sedimentation of large seminiferous tubule fragments. Supernatant was collected mixed with hypotonic extraction buffer (30 mM Tris HCl pH 8.2, 50 mM sucrose pH 8.2, 17 mM sodium citrate pH 8.2, 5 mM EDTA pH 8.2, 0.5 mM DTT) in 1:1 ratio and incubated for 5 minutes at room temperature. Cell suspensions were diluted three times in PBS pH 7.4 and

were centrifuged for 5 min at 1000 g. Supernatants were discarded and pelleted cells were resuspended in PBS pH 7.4. Cell suspensions were mixed 1:2 with 100 mM sucrose solution immediately prior to fixation. Cell suspensions were added to three to five times higher volume droplets of filtered (0.2 mm) 1% paraformaldehyde (PFA), 0.15% Triton X-100, 1 mM sodium borate pH 9.2 solution on diagnostic slides, and incubated for 70 min at room temperature in wet chambers. Then, slides were air dried either on the bench or in a fume hood. Finally, the slides were rinsed by water, incubated in 0.4% Photo-Flo 200 (Kodak) for 5-10 min, rinsed 3 times with distilled water, and dried at room temperature before proceeding to immunostaining.

**Preparation of oocyte spreads.** To prepare nuclear surface spread oocytes, two ovaries from each mouse were incubated in 20 μL hypotonic extraction buffer at room temperature for 15 min (Hypotonic Extraction Buffer/HEB: 30 mM Tris-HCl, 17 mM Trisodium citrate dihydrate, 5 mM EDTA, 50 mM sucrose, 0.5 mM DTT, 0.5 mM PMSF, 1x Protease Inhibitor Cocktail). After incubation, HEB solution was removed and 16 μL of 100 mM sucrose in 5 mM sodium borate buffer pH 8.5 was added. Ovaries were punctured by hypodermic needles (27Gx1/2") to release oocytes. Big pieces of tissue were removed. 9 μL of 65 mM sucrose in 5 mM sodium borate buffer pH 8.5 was added to the cell suspension and incubated for 3 min. After mixing, 1.5 μL of the cell suspension was added in a well containing 20 μL of fixative (1% paraformaldehyde, 1 mM borate buffer pH: 9.2, 0.15% Triton X-100) on a glass slide. Cells were fixed for 45 min in humid chambers, and then slides were air-dried. Upon completion of drying slides were washed with 0.4% Photo-Flo 200 solution (Kodak, MFR # 1464510) for 5 min and afterward they were rinsed with distilled water temperature before proceeding to immunostaining.

**Staining procedures of nuclear spreads.** Previously described blocking and immunostaining procedures[22] were optimized for immunostaining with each combination of antibodies; details are available upon request. In most cases, slides were treated with blocking buffer (2% BSA, 0.02% or 0.05% Na azide, 0.05% Tween20 and 0.05%Triton X-100, PBS, pH7.4) for 30 minutes before incubation with primary and secondary antibodies in blocking buffer. Combined immunostaining and PAR-FISH was performed on spermatocyte spreads as before[24].

**Immunofluorescence on gonad sections.** Testes were collected with care to ensure minimal disturbance of internal structure, and immediately fixed for 40 minutes at room temperature in 3.7% formaldehyde, 100 mM sodium phosphate pH 7.4, 0.1% Triton-X. After fixation, testes were washed 3 times in PBS and incubated in 30% sucrose, 0.02% sodium azide overnight at 4 °C. Afterwards, testes were frozen in OCT (Sakura Finetek Europe) on dry ice and stored at −80 °C until sectioned. 7 μM thick sections were cut with Leica CM1850, sections were dried on glass slides at room temperature. Slides were washed several times with distilled water to remove OCT, after which they were air dried. Dry slides were blocked with 2.5% w/v BSA in PBS for 1 hour, and then slides were stained as described for surface spreads.

To assess oocyte numbers in adult mice, DDX4 was detected in paraffin-embedded sections of ovaries in young adults (6 weeks old). Ovaries were dissected and fixed in 4% paraformaldehyde in 100 mM sodium phosphate buffer pH 7.4 overnight at 4 C. Afterward, ovaries were washed 3 times in PBS pH 7.4, once with 70% ethanol and embedded in paraffin for sectioning at 5 mm thickness. Deparaffinization and rehydration of the sections was performed as follows: 2×5 min in xylene, 2×5 min in 100% ethanol, 5 min each in 95%, 85%, 70%, 50% ethanol, 2×5 min in water. Sections were subjected to heat-mediated antigen retrieval in 10 mM Sodium citrate, 0.05% Tween 20, pH 6.0 for 20 min in boiling water bath. Sections were permeabilized in

PBS with 0.2% Triton X-100 for 45 min at room temperature and processed for immunofluorescence staining immediately. DDX4-positive oocytes were counted on every seventh section of both ovaries in each female mouse.

**Antibodies used for immunofluorescence and western blots.** The previously published antibodies[22,24,40,95,96] and commercial antibodies that were used in this study are listed in the reagent section of the supplementary information.

**Staging of meiotic prophase in nuclear spreads**
The first meiotic prophase substages were identified by a combination of three markers, SYCP3 (chromosome axis marker), SYCP1 (SC marker) and Histone H1t (post-mid pachytene marker in spermatocytes) (for details see also[24]). We defined preleptotene as a stage where hazy/punctate staining pattern of SYCP3 is observed throughout the nucleus as described previously[22]. The next stage, leptotene, is characterized by short stretches of axes and without any SC. This is a stage where recombination is initiated in wild type. The next stage is early zygotene, which is characterized by long but fragmented and incomplete axis stretches. In SC proficient genotypes, SC is also detected in this stage. Advancing to late zygotene stage, axes of all chromosomes are fully formed but SCs are incomplete. Cells enter pachytene when SC formation is completed in wild type. Whereas all chromosomes are fully synapse in pachytene oocytes, only autosomes synapse fully and heterologous sex chromosomes synapse only in their PARs in spermatocytes. SC formation was occasionally defective in *Iho1^{C7Δ/C7Δ}* spermatocytes; hence we applied modified criteria for pachytene stage to include cells that have synaptic defects. Prophase stage was considered pachytene in spermatocytes if axes fully formed, axes had condensed appearance typical of pachytene, and the status of SC satisfied one of the following criteria: i) if PARs of sex chromosomes were synapsed ii) if all autosomes were fully synapsed but PARs of sex chromosomes were unsynapsed iii) if PAR asynapsis was accompanied with partial or full asynapsis of up to 3 autosome pairs. Histone H1t staining was used to sub-stage mid- or late- pachytene. Histone H1t staining is absent or weak in early pachytene but intermediate and high Histone H1t levels are seen in mid and late pachytene, respectively. In diplotene stage, axes desynapse and the SC becomes fragmented. Histone H1t levels remain high in this stage in spermatocytes. In oocytes, the same stages exist but due to lack of H1t, the developmental time of fetuses was used to aid staging as before[64]. Most oocytes are in zygotene and mid-pachytene in fetuses 16 and 18 days postcoitum.

**Staging of mouse seminiferous tubule cross sections**
Coordinated mitotic proliferation, meiosis entry and migration of spermatogenic cells to the lumen of seminiferous tubules result in 12 well-defined stages of the seminiferous epithelial cycle (I-XII) that are characterized by distinct associations of premeiotic, meiotic and postmeiotic spermatogenic cell layers across cross sections of seminiferous tubules[97].

We used DAPI in combination with histone-H1t (marker of spermatocytes after mid pachytene or stages V-XII) to stage the epithelial cycle as before[24,98]. Cleaved-PARP was used to label and quantify apoptotic spermatocytes.

**Testis organ culture**
Organ cultures of testes were carried out as described earlier[34,99] with minor modifications. Testes of 8 dpp mice were dissected, tunica albuginea were removed, and each testis was cut into 2 or 3 pieces. Freshly isolated tubules were cultured at gas/liquid interphase on agarose gel blocks (1.5%; W/V; thickness about 7 mm). Gel pieces were preincubated in culture medium for 24 hours to saturate agarose with medium before samples were placed on them. After the biological

samples were placed on the agarose slices, the medium level was adjusted not to cover the seminiferous tubules in the culture wells. We used α-MEM (Life Technologies), 10% (v/v) KSR (Life Technologies) and gentamycin (Sigma-Aldrich) at a final concentration of 5 µg/ml as culture medium. The culture medium was supplemented with CDC7 inhibitors at concentrations indicated in the figure legends. We used various combinations of three inhibitors: TAK-931, XL413 hydrochloride and LY3143921 hydrate. Stock solutions of inhibitors were prepared per manufacturer's instructions. Tubules were incubated at 34 °C in humidified atmosphere containing 5% $CO_2$. After 2 days of incubation, samples were either processed for nuclear surface spreads or frozen for WB analysis.

## Multiple sequence alignments

Multiple sequence alignments were carried out using Tubingen Bioinformatic Toolkit[100,101] and Clustal Omega from the webserver of EMBL-EBI[102,103]. Jalview version 2 was used to edit alignments[104].

## Image analysis and quantification

Immunofluorescence images were captured by a Zeiss Axiophot upright microscope using Axiovision version 4.8 software or a Zeiss Axioimager.Z2 microscope using Zen2.3 Pro software.

Fiji variant of ImageJ 1.54 f was used to manually quantify Western blot band intensities[105–107]. Background of each lane was subtracted and signal was normalized to loading control of the same lane.

To reliably detect functional DSB-machinery clusters, we identified MEI4 foci (guinea pig-MEI4 antibodies) that colocalized with REC11 foci (rabbit-REC114 antibodies)[22]. Focus numbers of DSB factors (MEI4 and REC114) were quantified by Cell Profiler 3.0 software[108]. Macros of the Cell Profiler analysis are accessible at https://github.com/IhsanDereli/Iho1_C7del. A difference of gaussian blurs was used to define the borders of each MEI4 and REC114 focus similar to a previously used approach for the identification of DSB-factor foci[23] except that instead of automatic thresholding we empirically selected thresholds that allowed correct identification of DSB-clusters according to manual inspection in all of the examined meiotic stages for each experimental set. We considered a MEI4 focus as a functional DSB-machinery cluster only if at least 50% of the area of the MEI4 focus overlapped with a REC114 focus. The original grayscale images were masked based on identified foci and integrated signal intensity of MEI4 was calculated on original unprocessed images, after corrected for varying background signal by subtracting median of all pixel intensity values in the image. To compare focus numbers and intensities in distinct genotypes, we compared images of nuclear spreads that were stained in parallel within experimental repeats to limit technical variability. Furthermore, in each experimental set, we compared images from samples that originated from mice of similar ages (littermates when possible).

ssDNA associated protein foci were counted manually. ImageJ was used to process images for DMC1-RPA2 density measurements. First, we identified nuclear spreads where it was possible to delineate individual axis and SC stretches. Synapsis was identified by the presence of SYCP1 or depletion of HORMAD1. 'Segmented line' tool was used to mark and measure the length of synapsed and unsynapsed regions. DMC1 and RPA2 focus numbers were manually counted on synapsed or unsynapsed regions and DMC1 and RPA2 focus numbers were added up to estimate recombination focus numbers. If a DMC1 and an RPA2 focus overlapped or touched each other, we considered them as a single recombination focus if the center of one of the foci overlapped with the signal of the other focus. Recombination focus densities were calculated by dividing the sum of DMC1 and RPA2 foci in synapsed or unsynapsed regions with the total length of synapsed or unsynapsed chromosome axis, respectively, in each cell.

Images for figure panels were prepared with Adobe Photoshop CC19.

## Reagents and resources

Reagents, resources and softwares that were used in the manuscript are summarized in Supplementary Table. 7.

## Biological materials availability

Transgenic mouse strains and plasmids produced in this study are available from the authors upon request.

## Statistics and Reproducibility

Graphs were plotted by Graphpad Prism 9. Slopes and P values of regression lines were calculated by Graphpad Prism 9 in Fig. 6D–G. All other statistical tests were done using R version 4.1.3[109]. The statistical analyses were implemented in the R packages lmerTest and lme4[110,111]. The R code was published previously[98]. We used two-tailed statistics if this option was available in the statistical tests (Mann Whitney U-test and t tests but not the Likelihood ratio test[98]). There were no adjustments for multiple comparisons.

Specific randomization methods were not used. The study relied on comparison of wild type and various mutant mice that were generated by random segregation of alleles during sexual reproduction. Where control versus mutant mice were compared, samples were processed in parallel to eliminate batch effects. No formal sample-size calculations were performed and investigators were not blinded to the origin of samples during the assessment of results. All phenotypes were observed in at least two animals of each genotype and quantifications represent analysis of at least two independent animals. Sample sizes, statistical tests, and p-values are indicated in the figure legends and the source data file.

Reproducibility of representative experiments: images of immunoprecipitations, immunoblots, radiographs, immunofluorescence in histological sections, nuclear surface-spread meiocytes or oocytes and Y2H assays represent at least two independent (biological) repetitions of experiments.

## Reporting summary

Further information on research design is available in the Nature Portfolio Reporting Summary linked to this article.

## Data availability

The authors declare that the data supporting the findings of this study are available within the paper and its Supplementary Information. The mass spectrometry proteomics data generated in this study have been deposited in the ProteomeXchange Consortium via the PRIDE[112] partner repository with the dataset identifiers PXD042179 and PXD042221. All other data supporting the findings of this study are available from the corresponding author upon request. Source data are provided with this paper.

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

## Acknowledgements

We thank Dunja Knapp for insightful comments and proofreading the manuscript, M. Munzig for lab support, C. Santocanale for advice with DDK inhibitors and DDK assays, R. Jessberger for departmental support and sharing ideas and reagents (anti-SYCP3 antibody), E.L. Huttlin and S.P. Gygi for sharing mass spectrometry data and information about IHO1 phosphorylation, B. De-Massy, M. Biot and C. Grey for sharing ideas and unpublished data about genomic localization of DSB factors. I.D., V.T., F.P., K.R., M.S., N.S.I. and A.T. were supported by the Deutsche Forschungsgemeinschaft (DFG; grants: TO421/5-1, TO421/6-1/2, TO421/7-1, TO421/8-1/2, TO421/10-1, TO421/11-1, TO421/12-1 and TO421/14-1), HFSP research grant RGP0008/2015 and core funding from the Faculty of Medicine at the TU Dresden. Work in the S.K. laboratory was supported in part by US National Cancer Institute Cancer Center support grant P30 CA08748 and US National Institutes of Health grant R35 GM118092. J.R.W. and E.S. were funded by the Max Planck Society, and the DFG grant number WE 6513/2-1. B.N. and F.H. were funded by the GFF NÖ (Stiftungsprofessur, Province of Lower Austria) and the FFG (COIN Aufbau, Austrian Research Promotion Agency). We also acknowledge Grant "Gruppi di Ricerca 2020" from Regione Lazio, Italy (n. A0375-2020-36618) to MB, and Fondo di Beneficienza Intesa Sanpaolo (n. B/2021/0228) to MB. The Core Facility was generously supported by grants from European Regional Development Fund (ERDF/EFRE) (Contract #100232736) and the Deutsche Forschungsgemeinschaft (DFG) grants (INST 269/731-1 FUGG).

## Author contributions

I.D.: conceptualization, methodology, investigation, formal analysis, visualization, writing–original draft preparation, data curation and project administration. V.T., F.P., K.R., J.X., M.Bo., B.N., N.S.I., E.S., H.T., S.D., T.G., M.G., A.B.: investigation, formal analysis. M.Sta.: conceptualization, investigation, formal analysis. M.Ste.: supervision, writing–review and editing. M.Ba., A.S., J.R.W., F.H., S.K.: supervision, writing–review and editing, funding acquisition. A.T.: conceptualization, writing–original draft preparation, review and editing, supervision, project administration, funding acquisition.

## Funding

## Competing interests

The authors declare no competing interests.

## Additional information

[1]Institute of Physiological Chemistry, Faculty of Medicine at the TU Dresden, Fiedlerstrasse 42, 01307 Dresden, Germany. [2]Molecular Biology Program, Memorial Sloan Kettering Cancer Center, New York, NY 10065, USA. [3]Weill Cornell Graduate School of Medical Sciences, New York, NY 10065, USA. [4]Institute Krems Bioanalytics, IMC University of Applied Sciences, 3500 Krems, Austria. [5]Friedrich Miescher Laboratory of the Max Planck Society, Max-Planck-Ring 9, 72076 Tübingen, Germany. [6]Department of Developmental Biology, University of Hamburg, 22609 Hamburg, Germany. [7]University of Rome "Tor Vergata", Section of Anatomy, Via Montpellier, 1, 00133 Rome, Italy. [8]Core Facility Mass Spectrometry & Proteomics, Center for Molecular and Cellular Bioengineering (CMCB), Technische Universität Dresden, Dresden, Germany. [9]Saint Camillus International University of Health Sciences, Rome, Italy. [10]Howard Hughes Medical Institute, Memorial Sloan Kettering Cancer Center, New York, NY 10065, USA. ✉e-mail: attila.toth@mailbox.tu-dresden.de

