## [Peer Review File · Nature Communications]

Seeding the meiotic DNA break machinery and initiating recombination on chromosome axesREVIEWER COMMENTS

Reviewer #1 (Remarks to the Author):

The authors demonstrated that C-terminal acidic patch of IHO1 is essential for interacting with HORMAD1, and its interaction is regulated via phosphorylation(s) by DDK. The authors showed IHO1-HORMAD1 interaction ensures seeding of the DSB-machinery on axes for efficient pairing of homologous chromosomes. Further, they showed that without IHO1-HORMAD1 interaction, residual DSBs depend on ANKRD31, which enhances both the seeding and the growth of DSB-machinery clusters, highlighting complementary pathways that support seeding and growth of DSB-machinery clusters for recombination initiation in mice. The authors' comprehensive analyses using several mutant mice lines revealed the epistatic relationships among IHO1, HORMAD1, MEI4/REC114 and ANKRD31 in spatiotemporal pre-DSB cluster formation. and added more insights into the mechanisms of DSB formation together with the authors' previous studies on the pre-DSB meiotic recombination factors. Main message through the manuscript is clear. Overall manuscript is well organized, logical and thought-provoking. and raises several insights into meiotic DSB initiation in mice. Therefore, the manuscript should be open for the field. However, this reviewer raised several minor concerns as described below. Since this manuscript requires many knowledges of the authors' previous studies to fully understand their new findings, it would be tough for general readers to follow some of the presented data and interpretations of authors. Thus, this reviewer feels that most of them are solved by rewriting or rephrasing, but it would be better to add more textual descriptions in main text and graphical presentation.

Line 64-99: In the introduction, the authors summarized current knowledge on the pre-DSB meiotic recombination factors. But this reviewer feels that it could be difficult for general reader to follow detailed roles of those players. Also, questions to address in this study seems to be unclear due to complicated information. This reviewer suggests including a graphic summary somewhere which may help general readers to catch-up molecular interactions and roles of players IHO1, HORMAD1, MEI4/REC114, ANKRD31 and axis.

Line 231: Iho1C7Δ/C7Δ male mice were fertile despite meiotic apoptosis takes place. How does apoptosis affect spermatogenesis in Iho1C7Δ/C7Δ? Does apoptosis in Iho1C7Δ/C7Δ result in massive germ cell loss or do only a small population of spermatocytes die? Please add HE staining or images taken by lower magnification. It is helpful to understand how loss of IHO1-HORMAD1 interaction raised an impact on spermatogenesis.

The authors mentioned about mild defect in Iho1dC/dC females. It would be better to see whether there are univalents by metaphase I chromosome spreads.

Line 222-223, Fig. S2E-G:

The authors stated that DDK inhibition lowered recombination focus numbers, as detected by DMC1 staining, in both wild-type and *Iho1C7Δ/C7Δ* spermatocytes (Fig. S2E-G). Although the authors classified the cells by “IHO1 remains” and “IHO1 reduced”, this reviewer wonders whether they counted DMC1 foci in leptotene or zygotene spermatocytes, or whole early meiotic prophase cells including both. Since the number of DMC1 foci changes across the different stages of meiotic prophase even in the *Iho1C7Δ/C7Δ* spermatocytes (Fig.4F), this reviewer wonders what is the criteria of choosing spermatocytes for DMC1 foci counting. Please specify which stage of cells for data collection.

Line 274-277, Fig.3G

The authors stated that SmDSB-factor clusters were fewer in number but had consistently higher median signal intensities in *Iho1C7Δ/C7Δ Spo11-/-* and *Hormad1-/- Spo11-/-* spermatocytes as compared to *Spo11-/-* spermatocytes in all prophase stages that were present in these genotypes. The authors' careful assessment led to a conclusion that HORMAD1 and IHO1 play a role in resistance to destabilization by DSBs.

This reviewer assumed that in Fig.3G, *Spo11-/- Iho1+/+* and *Spo11-/- Hormad1+/+* were used as the controls for the litter mates of *Spo11-/- Iho1-/-* and *Spo11-/- Hormad1-/-*, respectively. *Spo11-/- Iho1+/+* and *Spo11-/- Hormad1+/+* mice should be the genetically same control in terms of *Spo11-/-* background. However, this reviewer wonders whether the signal intensity of SmDSB-factor clusters in immunostaining differs between *Spo11-/- Iho1+/+* and *Spo11-/- Hormad1+/+* at pre-leptotene. If so, this reviewer wonders how such subtle differences in the number and signal intensity of SmDSB-factor clusters are biologically significant. It would be better to explain that those measurements represent more than technical deviations.

Line 314-319: The authors stated “HORMAD1 ortholog Hop1 not only promotes DSBs, but also enables normal DSB repair kinetics and enhances the use of homologs instead of sister chromatids as recombination partners”. Do the authors want to point out a possibility of that DSB repair was completed via inter-sister chromatids happen in *Hormad1-KO* thereby number of recombination focus is lower? If so, please clearly state that possibility. This is very important because the authors are clamming that IHO1-HORMAD1 interaction is involved in efficient DSB formation.

Line 328-337: The authors carefully examined extent of synapsis according to the classification of axial morphology. Please show representative images of Fig. 5A (doty/short/log/full).

Line 338-339: The authors stated that both long and short autosomes were prone

339 to asynapsis in early pachytene. In Fig. 5C, there is an obvious cluster of asynaptic autosomes with middle length as well in early pachytene. For this reviewer, it seems that there is not such a bias in

asynaptic autosomes that derives from axis length. Please explain why the authors focused on long and short autosomes in main text.

Line 339: The authors mentioned about apoptosis triggered by asynaptic (long) chromosomes. In Fig. 5B, late pachytene spermatocytes carrying fully unsynapsed autosomes were observed in *Iho1C7Δ/C7Δ*, which means those spermatocytes could be survivors from pachytene checkpoint. Were those chromosomes in such survivor short? If the authors can present distribution of chromosomes size carrying fully unsynapsed autosomes or show representative image, it further supports the idea described line 341-343.

Line 352: Meaning of “12 autosomes and 25 sex chromosome pairs in spermatocytes, 32 chromosomes in oocytes” is unclear. Please describe how many of spermatocytes/oocytes the authors totally counted.

Line 431: Grammar error (axes, Accordingly, ANKRD31)

Reviewer #2 (Remarks to the Author):

In this study Dereli et al. discovered that efficient biogenesis of the meiotic DSB-machinery in mice requires seeding by axial IHO1 platforms, enabling downstream clustering of DSB factors. They propose that this function of IHO1 is controlled by the DBF4-dependent kinase (DDK), which modulates the interaction between IHO1 and the chromosomal axis component HORMAD1. The authors also find that in the absence of the IHO1-HORMAD1 interaction, residual DSBs depend on ANKRD31, which enhances both the seeding and the growth of DSB-machinery clusters. Thus, in their model, recombination initiation is ensured by complementary pathways that either support ‘seeding’ or ‘growth’ of DSB-machinery clusters. Together, seeding and growth synergistically enable DSB machinery condensation on chromosomal axes.

In my opinion, the work is of excellent quality and the data supports the main conclusions made. I also think that the results are of interest to the meiosis community and beyond. I would suggest that the authors address the following minor points, ordered as they appear in the manuscript and not by relevance:

Comments:

1. Perhaps the authors could change the first sentence in the abstract. It is very long and it also disregards that programmed DNA DSBs are generated during the assembly and diversification of lymphocyte antigen receptor genes (i.e. they are not necessarily a 'unique meiotic feature').

2. The sentence in lines 102-105 of the intro needs revision.

3. In lines 161-164 the authors write "Whereas S569 and S570 seem to have a critical function, we are cautious with interpreting the phenotypes caused by non phosphorylatable and phosphomimetic replacements at these sites. Hence, the significance of in vivo phosphorylation at S569 and S570 remains uncertain." I compliment the authors for being cautious here. However, together with the Y2H data showing that the interaction in yeast most likely occurs without phosphorylation (unless yeast modifies exactly those residues), and the in vitro data in Fig S1B, it is rather likely that Serines, rather than phosphorylation, are needed for the interaction. Does phosphorylation stimulate the interaction in vitro? In the HORMAD1 IPs, in Figure 1E, is the co-IPed IHO1 in the phosphorylated state? The electrophoretic mobility in that gel is hard to interpret as the input IHO1 is not running in two forms, as in other gels.

4. In lines 202-206 the authors write: "Alternatively, in the absence of S569/570, phosphorylation of S476 and T490 or other unknown sites were insufficient for a discernible mobility shift in standard SDS-PAGE. The latter hypothesis was supported by analysis on phos-tag gels (Fig. S2B), which enable detection of distinct phosphoforms of proteins by enhancing their retardation during electrophoresis". To make this point better, the authors should treat the IHO1C7Δ with phosphatase. Even though unlikely, it is formally possible that the band that is left in the mutant is the phosphorylated version, instead of the unphosphorylated.

5. In lines 227-229: 'Thus, reliance of recombination initiation on DDK is conserved between mice and yeast, with a divergence in the nature of DDK-controlled functions.' I would recommend that the authors are even more careful here. They have not necessarily analysed the process in a fully comparable manner to what has been done in yeast. For example, the localization of fully validated phosphorylation-resistant IHO1 mutants is missing in mice.

6. To complement the experiments with DDK inhibition, it would have been ideal to perform coIP experiments between IHO1 and its interactors. I am not sure if this is possible, given the amounts of DDK inhibitor required. If not possible, claiming in the abstract that DDK phosphorylates the c-term of IHO1 to promote interaction with HORMAD1 is a bit too strong, in my opinion. Could the authors test at least in vitro if DDK phosphorylation stimulates the interaction?

7. The discussion is overall nice and thought provoking. However, I think it would have been useful to discuss the involvement of DDK, and more generally phosphorylation, which are not at all mentioned. What might phosphorylation do?

Reviewer #3 (Remarks to the Author):

Overall, the manuscript is well designed with contribution to the field of DSB-machinery formation during meiosis. The study found that IHO1 C-terminus is essential for IHO1-HORMAD1 interaction, IHO1-axis association, DSB-factors recruitment on chromosome axes and DSB formation. The function of IHO1 C-terminus might be mediated by phosphorylation on S560 and S570. The scientific question of the manuscript is clear and the data are presented in a logical manner. However, some concerns shown below are suggested to be addressed.

Comments:

1. Line 119-120, the Y2H assay were conducted between different truncated IHO1 and HORMAD1. The interactions were not narrowed down to the HORMA domain. It is unappropriated to present "IHO1 fragments that included the C-terminal 75 amino acids of IHO1 interacted with HORMAD1 and, specifically, its HORMA domain".
2. In Fig 1B, what does "cross" mean? Does the cross indicate that the corresponding Y2H experiments were not conducted? If so, why the Y2H experiments corresponding to these "crosses" were not conducted? On the contrary, there are no "cross" shown in Fig. S2C. Please show the data in a consistent way.
3. In Fig 1B, the panel and legend does not match.
4. Line126-130, it is better to confirm the relationship between HORMAD1 and IHO1_WT, IHO1_S-T_AA or IHO1_A-A_SS using eukaryotic cell system, to verify the importance of S569 and S570 in mediating the interaction between HORMAD1 and IHO1.

5. Line189, it presented that phosphorylation of the IHO1 C-terminus enhances axial accumulation. Axial accumulation of IHO1 relies on the interaction between IHO1 and HORMAD1. Whether the interaction between IHO1 and HORMAD1 is affected by phosphorylation on S569 and/or S570 should be confirmed.

6. Deletion of C-terminus of IHO1 affected the recruitment of MEI4 and REC114 to chromosome axis. Does the deletion of C-terminus of IHO1 affect the protein expression level of MEI4 and REC114? Can IHO1 C7Δ/C7Δ form complex with MEI4 and REC114 outside the chromosome axis?

7. Fig. 5G, Fig. S3A-B, Fig. 3E-F, Fig. 4I-J, all the assays were conducted in two independent experiments. In each assay, how many mice were analyzed?

8. Line 212-213, Should “Fig 2” be “Fig 2D”?

9. Some formats should be consistent throughout the manuscript. For instance, “2 fold” 、 “twofold” or “2-fold”?

10. Line 885, 913, 1027, the text should be bold.

11. The Results section is relatively verbose with lots of discussions. For instance, it might be better to move line 363-373 to the Discussion section.

We thank the reviewers for their efforts and their constructive suggestions. Please, find below our point-by-point responses to reviewer comments, manuscript NCOMMS-23-34658. Please, note that we submitted two versions of the manuscript that differ in the formatting of tracked changes. In the fully tracked (1st) version, we show insertions in line (marked red) and deletions in balloons on the side. The alternative (2nd) version highlights insertions in red without showing the deleted text to enhance readability. In our response, we describe changes by referring to line numbers in the simple (2nd) version that hides deletions and lacks balloons.

REVIEWER COMMENTS

Reviewer #1 (Remarks to the Author):

The authors demonstrated that C-terminal acidic patch of IHO1 is essential for interacting with HORMAD1, and its interaction is regulated via phosphorylation(s) by DDK. The authors showed IHO1-HORMAD1 interaction ensures seeding of the DSB-machinery on axes for efficient pairing of homologous chromosomes. Further, they showed that without IHO1-HORMAD1 interaction, residual DSBs depend on ANKRD31, which enhances both the seeding and the growth of DSB-machinery clusters, highlighting complementary pathways that support seeding and growth of DSB-machinery clusters for recombination initiation in mice. The authors' comprehensive analyses using several mutant mice lines revealed the epistatic relationships among IHO1, HORMAD1, MEI4/REC114 and ANKRD31 in spatiotemporal pre-DSB cluster formation. and added more insights into the mechanisms of DSB formation together with the authors' previous studies on the pre-DSB meiotic recombination factors. Main message through the manuscript is clear. Overall manuscript is well organized, logical and thought-provoking. and raises several insights into meiotic DSB initiation in mice. Therefore, the manuscript should be open for the field. However, this reviewer raised several minor concerns as described below. Since this manuscript requires many knowledges of the authors' previous studies to fully understand their new findings, it would be tough for general readers to follow some of the presented data and interpretations of authors. Thus, this reviewer feels that most of them are solved by rewriting or rephrasing, but it would be better to add more textual descriptions in main text and graphical presentation.

Response: We are glad that the reviewer liked our manuscript and found our data comprehensive and thought-provoking. We also appreciate the reviewer's insightful comments, which follow.

Remark

Line 64-99: In the introduction, the authors summarized current knowledge on the pre-DSB meiotic recombination factors. But this reviewer feels that it could be difficult for general reader to follow detailed roles of those players. Also, questions to address in this study seems to be unclear due to complicated information. This reviewer suggests including a graphic summary somewhere which may help general readers to catch-up molecular interactions and roles of players IHO1, HORMAD1, MEI4/REC114, ANKRD31 and axis.

Response: We agree that the background information is complicated, therefore we added a graphical overview of the previously reported knowledge regarding the composition and regulation of DSB factor clusters (Supplementary Fig. 1).

Remark

Line 231: *Iho1*^{C7Δ/C7Δ} male mice were fertile despite meiotic apoptosis takes place. How does apoptosis affect spermatogenesis in *Iho1*^{C7Δ/C7Δ}? Does apoptosis in *Iho1*^{C7Δ/C7Δ} result in massive germ cell loss or do only a small population of spermatocytes die? Please add HE staining or images taken by lower magnification. It is helpful to understand how loss of IHO1-HORMAD1 interaction raised an impact on spermatogenesis.

Response: We included tiled images of cleaved PARP and DAPI stained testis sections to show that most if not all tubules contain postmeiotic cells in *Iho1*^{C7Δ/C7Δ}, consistent with the moderate levels of apoptosis in spermatocytes. Quantification of apoptotic cells (Supplementary Table 5) also shows that most seminiferous tubules contain less than 5 apoptotic pachytene/diplotene/dividing meiocytes, which is a low number given that each tubule section tends to contain minimum 30-40 of these cell types.

Remark

The authors mentioned about mild defect in *Iho1*dC/dC females. It would be better to see whether there are univalents by metaphase I chromosome spreads.

Response: During our initial analysis, we also considered counting univalents (which reflect the absence of crossovers/chiasmata) on spreads of *in vitro* matured oocytes, to assess crossover/chiasma formation as part of our characterization of the female recombination defects in *Iho1*^{C7Δ/C7Δ}. However, after careful consideration, we decided against this analysis. This is because (1) absence of univalents is a downstream/indirect consequence of asynapsis and reduced recombination focus numbers, which are upstream defects we comprehensively quantified in *Iho1*^{C7Δ/C7Δ}, (2) quantification of univalents is work and time intensive, and (3), in our opinion, it would add very little to the main message of the manuscript or to the understanding of female fertility defects. Detailed explanation follows below.

The mild fertility defects of female *Iho1*^{C7Δ/C7Δ} mice are well explained by the recombination and SC defects which are reported in the manuscript. We show that DSB numbers are reduced (Supplementary Fig. 7C-H formerly 4C-H), which leads to an inability to complete SC formation in approximately half of *Iho1*^{C7Δ/C7Δ} oocytes (Supplementary Fig. 9D formerly 5D). It is known, and documented by multiple studies, that incomplete SC formation triggers oocyte apoptosis around birth in females. Hence, SC defects match the approximately 2-fold reduction of oocyte numbers in young *Iho1*^{C7Δ/C7Δ} females as compared to wild type (Supplementary Fig. 5D-E, formerly 3C-D). The observed level of oocyte number reduction is expected to have little or no influence on fertility in young *Iho1*^{C7Δ/C7Δ} females. But reduced oocyte numbers are expected to result in early depletion of oocytes during aging, leading to reduced litter sizes in old mice. This is exactly what we observed in *Iho1*^{C7Δ/C7Δ} mice (Supplementary Table 4) suggesting that SC defects and resultant reduction in oocyte numbers is the main reason for subfertility in female *Iho1*^{C7Δ/C7Δ} mice.

Furthermore, we quantified the numbers of MLH1 foci, which mark future crossovers. Importantly, MLH1 focus numbers are nearly normal in *Iho1*^{C7Δ/C7Δ} oocytes that succeed in SC formation (median 26 MLH1 foci in *Iho1*^{C7Δ/C7Δ} as compared to 27 in wild type, Supplementary Figure 9H formerly 5H), suggesting that crossover formation is barely affected in *Iho1*^{C7Δ/C7Δ}. Thus, quantifying univalent numbers to assess crossover formation would add very little new information to our analysis, which, in our opinion, does not justify the significant investment required for this experiment. We would

like to ask the reviewer to accept our well-considered decision on this matter, based on the above arguments.

Remark

Line 222-223, Fig. S2E-G:

The authors stated that DDK inhibition lowered recombination focus numbers, as detected by DMC1 staining, in both wild-type and *Iho1C7Δ/C7Δ* spermatocytes (Fig. S2E-G). Although the authors classified the cells by “IHO1 remains” and “IHO1 reduced”, this reviewer wonders whether they counted DMC1 foci in leptotene or zygotene spermatocytes, or whole early meiotic prophase cells including both. Since the number of DMC1 foci changes across the different stages of meiotic prophase even in the *Iho1C7Δ/C7Δ* spermatocytes (Fig.4F), this reviewer wonders what is the criteria of choosing spermatocytes for DMC1 foci counting. Please specify which stage of cells for data collection.

Response: We counted DMC1 focus numbers in early zygotene, as identified by axis morphology (long but incomplete axes). We added this detail to the result section (line 243 in revised manuscript).

Remark

Line 274-277, Fig.3G

The authors stated that SmDSB-factor clusters were fewer in number but had consistently higher median signal intensities in *Iho1C7Δ/C7Δ Spo11^{-/-}* and *Hormad1^{-/-} Spo11^{-/-}* spermatocytes as compared to *Spo11^{-/-}* spermatocytes in all prophase stages that were present in these genotypes. The authors' careful assessment led to a conclusion that HORMAD1 and IHO1 play a role in resistance to destabilization by DSBs.

This reviewer assumed that in Fig.3G, *Spo11^{-/-} Iho1^{+/+}* and *Spo11^{-/-} Hormad1^{+/+}* were used as the litter mates of *Spo11^{-/-} Iho1^{-/-}* and *Spo11^{-/-} Hormad1^{-/-}*, respectively. *Spo11^{-/-} Iho1^{+/+}* and *Spo11^{-/-} Hormad1^{+/+}* mice should be the genetically same control in terms of *Spo11^{-/-}* background. However, this reviewer wonders whether the signal intensity of SmDSB-factor clusters in immunostaining differs between *Spo11^{-/-} Iho1^{+/+}* and *Spo11^{-/-} Hormad1^{+/+}* at pre-leptotene. If so, this reviewer wonders how such subtle differences in the number and signal intensity of SmDSB-factor clusters are biologically significant. It would be better to explain that those measurements represent more than technical deviations.

Response: We did not carry out statistical comparison between *Spo11^{-/-} Iho1^{+/+}* and *Spo11^{-/-} Hormad1^{+/+}* mice because they originate from distinct colonies that were bred separately for considerable times. Samples were also processed in separate experiments in these two colonies. Due to considerable technical variation in chromosome spreading and staining we do not consider *Spo11^{-/-} Iho1^{+/+}* and *Spo11^{-/-} Hormad1^{+/+}* samples directly comparable. Whereas these datasets would be statistically different, we cannot tell if this difference was due to genetic drift between distinct mouse colonies or technical variation. We added further explanation to the figure legend and the methods (Figure legends, lines 723-726, image quantifications in methods, lines 1193-1197-y), to clarify what samples were statistically compared in our analyses.

Remark

Line 314-319: The authors stated “HORMAD1 ortholog Hop1 not only promotes DSBs, but also enables normal DSB repair kinetics and enhances the use of homologs instead of sister

chromatids as recombination partners”. Do the authors want to point out a possibility of that DSB repair was completed via inter-sister chromatids happen in Hormad1-KO thereby number of recombination focus is lower? If so, please clearly state that possibility. This is very important because the authors are claiming that IHO1-HORMAD1 interaction is involved in efficient DSB formation.

Response: We spelled out the possibility of inter sister DSB repair in Hormad1-/- mice as requested (see lines 330-333).

Remark

Line 328-337: The authors carefully examined extent of synapsis according to the classification of axial morphology. Please show representative images of Fig. 5A (doty/short/log/full).

Response: We included a new supplementary Fig. 8 showing the distinct categories of axis and SC development in WT and *Iho1*^{C7Δ/C7Δ} mice.

Remark

Line 338-339: The authors stated that both long and short autosomes were prone 339 to asynapsis in early pachytene. In Fig. 5C, there is an obvious cluster of asynaptic autosomes with middle length as well in early pachytene. For this reviewer, it seems that there is not such a bias in asynaptic autosomes that derives from axis length. Please explain why the authors focused on long and short autosomes in main text.

Response: We are sorry for the confusion. We intended to convey that autosomes of all lengths were affected by asynapsis in early pachytene. We categorized chromosome length as either short or long without distinguishing a middle range. We realize that this may be confusing, hence we rephrased the relevant sentence, and we refer to autosomes of all lengths in the revised text (see lines 362).

Remark

Line 339: The authors mentioned about apoptosis triggered by asynaptic (long) chromosomes. In Fig. 5B, late pachytene spermatocytes carrying fully unsynapsed autosomes were observed in *Iho1*^{C7Δ/C7Δ}, which means those spermatocytes could be survivors from pachytene checkpoint. Were those chromosomes in such survivor short? If the authors can present distribution of chromosomes size carrying fully unsynapsed autosomes or show representative image, it further supports the idea described line 341-343.

Response: We completely agree with the reviewer, and we wanted to convey the conclusion mentioned by the reviewer in lines 364-367 of the revised manuscript. In line with the suggestion of the reviewer we separately show the length-rank of partially asynaptic and fully unsynapsed autosomes in Fig. 5C of the revised manuscript. We note that, by mistake, we reported 75 asynaptic chromosomes in early pachytene instead of 73 (the latter is the correct number), and we showed data only from early and mid pachytene cells in the previous version of the manuscript. In the current version we corrected these mistakes, and we show early, mid and late pachytene stages.

Remark

Line 352: Meaning of “12 autosomes and 25 sex chromosome pairs in spermatocytes, 32 chromosomes in oocytes” is unclear. Please describe how many of spermatocytes/oocytes the authors totally counted.

Response: We added the number of spermatocytes and oocytes that were examined (lines 375-380). This is the same as in the corresponding Fig. 5 and Supplementary Fig 9 (formerly S5).

Remark

Line 431: Grammar error (axes, Accordingly, ANKRD31)

Response: Thank you for pointing out this typing error, we corrected it.

Reviewer #2 (Remarks to the Author):

In this study Dereli et al. discovered that efficient biogenesis of the meiotic DSB-machinery in mice requires seeding by axial IHO1 platforms, enabling downstream clustering of DSB factors. They propose that this function of IHO1 is controlled by the DBF4-dependent kinase (DDK), which modulates the interaction between IHO1 and the chromosomal axis component HORMAD1. The authors also find that in the absence of the IHO1-HORMAD1 interaction, residual DSBs depend on ANKRD31, which enhances both the seeding and the growth of DSB-machinery clusters. Thus, in their model, recombination initiation is ensured by complementary pathways that either support ‘seeding’ or ‘growth’ of DSB-machinery clusters. Together, seeding and growth synergistically enable DSB machinery condensation on chromosomal axes.

In my opinion, the work is of excellent quality and the data supports the main conclusions made. I also think that the results are of interest to the meiosis community and beyond. I would suggest that the authors address the following minor points, ordered as they appear in the manuscript and not by relevance:

Response: We are very glad that the reviewer judges our manuscript high quality, and we greatly appreciate his/her constructive comments that follow.

Comments:

Remark

1. Perhaps the authors could change the first sentence in the abstract. It is very long and it

also disregards that programmed DNA DSBs are generated during the assembly and diversification of lymphocyte antigen receptor genes (i.e. they are not necessarily a ‘unique meiotic feature’).

Response: We agree with the comment. Accordingly, we altered and split the first sentence of the abstract as requested (lines 30-32).

Remark

2. The sentence in lines 102-105 of the intro needs revision.

Response: We modified the start of the relevant paragraph including the indicated sentence (lines 104-109).

Remark

3. In lines 161-164 the authors write” “Whereas S569 and S570 seem to have a critical function, we are cautious with interpreting the phenotypes caused by non phosphorylatable and phosphomimetic replacements at these sites. Hence, the significance of *in vivo* phosphorylation at S569 and S570 remains uncertain.” . I compliment the authors for being cautious here. However, together with the Y2H data showing that the interaction in yeast most likely occurs without phosphorylation (unless yeast modifies exactly those residues), and the *in vitro* data in Fig S1B, it is rather likely that Serines, rather than phosphorylation, are needed for the interaction. Does phosphorylation stimulate the interaction *in vitro*? In the HORMAD1 IPs, in Figure 1E, is the co-IPed IHO1 in the phosphorylated state? The electrophoretic mobility is that gel is hard to interpret as the input IHO1 is not running in two forms, as in other gels.

Response (also relevant to points 4-7):

3.1 Explanation of our conclusions about the proposed roles of IHO1 phosphorylation

We agree with the reviewer that the data show that the presence of serines in positions 569 and 570 are important for IHO1-HORMAD1 interaction, and that the phosphorylation of these serines is not necessary for the interaction *in vitro*. Yet, *in vivo* data suggest that IHO1 phosphorylation enhances IHO1 recruitment to axis as detailed below.

The observation that replacement of serines with phosphomimetic aspartates or glutamates hinders IHO1-HORMAD1 interaction also raises the possibility that only unphosphorylated serines enable strong IHO1-HORMAD1 interaction. However, phosphorylated serines and phosphomimetic amino acids are chemically different. Hence, it is uncertain if phosphomimetic amino acid replacements correctly model the role of *in vivo* phosphorylation of IHO1 C-terminus.

Importantly, the slowest migrating form of IHO1 (which seem to represent IHO1 that is phosphorylated on its C-terminus) preferentially accumulates in the chromatin-rich insoluble fraction of testis extracts, showing that IHO1 phosphorylation correlates with chromatin binding (Fig. 2A and Supplementary Fig 3A-B). This suggests that phosphorylation of IHO1 C-terminus promotes chromatin binding of IHO1 and/or happens as a consequence of chromatin binding of IHO1. The slow migration form of IHO1 is also present in *Hormad1*^{-/-} mice (Fig. 2B) where IHO1 cannot efficiently bind axes/chromatin (Supplementary Fig 2H). This observation suggests (by way of exclusion) that phosphorylation of IHO1 C-terminus is not a consequence, but rather a promoter of IHO1 binding to HORMAD1 on chromosome axes *in vivo*. This conclusion is also supported by the observation that inhibition of DDK leads to both reduced IHO1 phosphorylation and reduced loading of IHO1 on chromosome axes *in vivo* (Fig. 2E-H). Altogether, our experiments suggest that phosphorylation of

IHO1 C-terminus enhances (but is probably not essential for) IHO1-HORMAD1 interaction and IHO1 axis binding *in vivo*.

We agree that the previous version of the manuscript did not sufficiently discuss limitations and contradicting features of *in vitro* and *in vivo* phosphorylation analysis. Therefore, we included additional reasoning in the Results and Discussion (lines 168-173, 222-226, 243-246 and 560-582) to provide a more nuanced interpretation of the data, highlighting the possibility that unphosphorylated serines in the C-terminus are important for IHO1-HORMAD1 interaction.

3.2 Response to Questions from point 3.:

“...Does phosphorylation stimulate the interaction *in vitro*? In the HORMAD1 IPs, in Figure 1E, is the co-IPed IHO1 in the phosphorylated state?...”

Despite investing lots of efforts into addressing the role of IHO1 phosphorylation *in vivo* over the last three-to-four years, we have not been able to provide definitive answers to the above questions, due to technical difficulties. A list of our approaches and the experienced difficulties follow:

- (a) We attempted to raise antibodies that recognize the unphosphorylated or the phosphorylated IHO1 C-terminus, but these antibodies were not specific, which precluded easy and definitive monitoring of C-terminal phosphorylation of IHO1 in cytological samples or protein extracts.
- (b) Phospho mapping by mass spectrometry of IPs has also been difficult as the C-terminal peptide appears underrepresented in these experiments due to poor flying properties of the peptide.
- (c) For an unknown reason, it is very difficult to resolve the low and high mobility forms of IHO1 in IPs. We repeated this experiment multiple times, and independent of how much IP products we loaded on gels, we were only able to detect a smeared IHO1 band that overlaps with both the slow and fast migrating bands of IHO1 in input samples. To resolve this issue, we also attempted analysis of IPs on Phos tag gels. However, Phos-tag gels are tricky to work with, and we found that the buffer and protein composition of extracts has a strong influence on the electrophoretic mobility of proteins in Phos-tag gels, which led to inconclusive band patterns in IHO1/HORMAD1 IPs.
- (d) It has been also difficult to conclusively address the role of phosphorylation *in vitro*. Only 10-11 percent of a short (15 amino acid) C-terminal peptide of IHO1 was phosphorylated by purified DDK, and only the serine corresponding to S570 was phosphorylated *in vitro* in these peptides (Fig. 2). This may show that (i) DDK collaborates with other kinase(s) in the phosphorylation of the C-terminus of IHO1, (ii) other factors that are present *in vivo* are needed for efficient IHO1 phosphorylation by DDK, and/or (iii) only longer IHO1 fragments/full length IHO1 is phosphorylated efficiently by DDK. Prompted by the reviewers question, we also attempted to perform *in vitro* pull down assays with longer IHO1 fragments (359- 574 and 440-574 amino acid positions) following phosphorylation with purified DDK. However, DDK treatment did not lead to efficient retardation of electrophoretic mobility of IHO1 fragments under the conditions we tried. This indicated that either DDK inefficiently phosphorylated the examined IHO1 fragments, and/or DDK phosphorylation did not lead to appreciable mobility shift in the absence of additional phosphorylation by other kinases. Thus, answering if phosphorylation alters *in vitro* binding affinity of IHO1 fragments to HORMAD1 will require lengthy optimization that will take an extended period of time and have uncertain outcomes. In general, the *in vitro* IHO1-HORMAD1 binding assays have considerable limitations and will require complicated controls. Neither us nor other groups (Laroussi H,Kadlec J, 2023 EMBO J, doi: 10.15252/embj.2023113866. Epub 2023 Jul 11.) have succeeded in purifying full length IHO1. This poses limitation in addressing the role of IHO1 phosphorylation *in vitro*:
The C-terminal 7 amino acids of IHO1 (including S569 and S570) may or may not be the primary interface between IHO1 and HORMAD1 *in vivo*. In the later scenario, the wild-type C-terminus may support an IHO1 conformation that permits the interaction of other (non-C-terminal) IHO1 region(s) with HORMAD1 (a similar/analog scenario was reported for the budding yeast orthologs of IHO1 and HORMAD1 complexes, Rousova D,Weir J, 2021, eLife, doi: 10.7554/eLife.72330). It follows that we may need to study the interaction of full length IHO1 and HORMAD1 to understand the effect of phosphorylation of the IHO1 C-terminus. A further complication is that, *in vivo*, IHO1 seems to primarily interact with chromosome axis-bound fraction of HORMAD1, which is thought to have

different conformation than the soluble fraction of HORMAD1 (note that IHO1-HORMAD1 interaction/co-localization requires the chromosome axis protein SYCP2 according to Fujiwara Y, ... Ishiguro KI, 2020, PLoS Genetics, doi: 10.1371/journal.pgen.1009048.). Therefore, it is uncertain, if (and to what extent) *in vitro* pull-down assays of IHO1 fragments and stand-alone HORMAD1 are able to model the effect of IHO1 phosphorylation *in vivo*. A more valid *in vitro* analysis of phosphorylation will require complex experiments that involve co-expression of axis proteins in addition to IHO1 and HORMAD1.

Due to their complexity, these *in vitro* experiments will take very long time, have uncertain outcomes, and their goals go well beyond the scope of the current manuscript. Based on the above listed arguments, we kindly ask you and the reviewer to accept our decision not to carry out further *in vitro* experiments to test the role of phosphorylation of IHO1 C-terminus in IHO1-HORMAD1 interaction.

3.3

Summary of response to point 3: To address the reviewer's comments we added additional explanations to the Results and the Discussion. In the revised manuscript, we point out the uncertainties regarding the roles of IHO1 phosphorylation and DDK (lines 168-173, 222-226, 243-246 and 560-582). We note that, in our opinion, these uncertainties do not have a major influence on the overall importance of the manuscript, because they do not affect the main message of the manuscript. Specifically, our core message is that chromosome axis-associated seeding and growth of DSB factor clusters are genetically separable processes that depend on IHO1-HORMAD1-mediated and ANKRD31-mediated pathways, respectively.

Remark

4. In lines 202-206 the authors write: "Alternatively, in the absence of S569/570, phosphorylation of S476 and T490 or other unknown sites were insufficient for a discernible mobility shift in standard SDS-PAGE. The latter hypothesis was supported by analysis on phos-tag gels (Fig. S2B), which enable detection of distinct phosphoforms of proteins by enhancing their retardation during electrophoresis". To make this point better, the authors should treat the IHO1C7Δ with phosphatase. Even though unlikely, it is formally possible that the band that is left in the mutant is the phosphorylated version, instead of the unphosphorylated.

Response: We performed the requested experiment, and we also rephrased the interpretations of the dephosphorylation experiments (lines 214-222) which is shown below for your convenience:

"Thus, loss of the last 7 amino acids, and the S569/570 phospho-sites within, may have prevented phosphorylation of all sites in IHO1. Alternatively, IHO1_C7Δ is phosphorylated on S476 and T490 or other unknown sites but without resulting in a strong mobility shift in standard SDS-PAGE. The latter hypothesis was supported by analysis on phos-tag gels (Fig. Supplementary Fig. 3B), which enable detection of distinct phosphoforms of proteins by enhancing their retardation during electrophoresis^{45, 46}. In addition, dephosphorylation induced a slight increase in the electrophoretic mobility of IHO1_C7Δ in standard SDS-PAGE, consistent with residual phosphorylation of unknown sites in IHO1_C7Δ (Supplementary Fig. 3C)."

Importantly, the newly included experiment supports our original conclusion that loss of the last seven amino acids does not eliminate phosphorylation of IHO1 at sites outside of the IHO1 C-terminus.

Remark

5. In lines 227-229: ‘Thus, reliance of recombination initiation on DDK is conserved between mice and yeast, with a divergence in the nature of DDK-controlled functions.’ I would recommend that the authors are even more careful here. They have not necessarily analysed the process in a fully comparable manner to what has been done in yeast. For example, the localization of fully validated phosphorylation-resistant IHO1 mutants is missing in mice.

Response: We agree that the yeast and mouse data is not fully comparable, and that the role of DDK is less well understood in mice. Accordingly, we toned down the statement about potential conservation of DDK functions and moved this section to the Discussion (lines 570-582, as well as addressing the limitations of the conclusions in lines 576-582). Nevertheless, our data suggest that DDK (directly or indirectly) enhances both recombination initiation and IHO1 accumulation on chromosome axes in mice. Whereas DDK has a well-documented role in initiation of DSBs in budding yeast, it is not required for chromatin/axis association of the budding yeast ortholog of IHO1. Hence, we think it is justified to cautiously state that there seem to be both conservation and divergence in the role of DDK in meiotic recombination initiation.

Remark

6. To complement the experiments with DDK inhibition, it would have been ideal to perform coIP experiments between IHO1 and its interactors. I am not sure if this is possible, given the amounts of DDK inhibitor required. If not possible, claiming in the abstract that DDK phosphorylates the c-term of IHO1 to promote interaction with HORMAD1 is a bit too strong, in my opinion. Could the authors test at least *in vitro* if DDK phosphorylation stimulates the interaction?

Response: We agree that coIPs of IHO1 would be technically demanding in samples where DDK inhibition was performed due to the small sample sizes used in these experiments. Furthermore, interpretation of coIPs would likely be ambiguous, due to partial effects on IHO1 phosphorylation/localization in samples exposed to DDK inhibitors (50-60% of IHO1 shifts to the fast migrating form, and IHO1 is depleted (but not fully removed) from axis in 50-70% of spermatocytes, Fig. 2F-H). Therefore, we carried out analysis of IHO1 localization in single cells instead of analysis of bulk tissue by IPs.

Addressing if DDK phosphorylation stimulates the IHO1-HORMAD1 interaction *in vitro* also have considerable difficulties and limitations, which prevents(ed) us from performing the suggested *in vitro* experiment (please, see point 3.2d in the response to the 3rd remark of reviewer 2 for a detailed explanation).

Therefore, to satisfy the request made in this point, we toned down the description of the DDK results in the abstract, avoiding definitive statements about the potential role of DDK in IHO1 phosphorylation (lines 36-38).

Remark

7. The discussion is overall nice and thought provoking. However, I think it would have been useful to discuss the involvement of DDK, and more generally phosphorylation, which are not at all mentioned. What might phosphorylation do?

Response: We agree with the reviewer that an extended description of our hypotheses about the roles of IHO1 phosphorylation and DDK benefits the manuscript. Therefore, we included additional explanation/discussion of experiments concerning IHO1 phosphorylation in both the Results and the Discussion sections of the revised manuscript (lines 168-174, 222-226, 243-246 and 560-582, and response to remarks 3-6 of reviewer 2).

Reviewer #3 (Remarks to the Author):

Overall, the manuscript is well designed with contribution to the field of DSB-machinery formation during meiosis. The study found that IHO1 C-terminus is essential for IHO1-HORMAD1 interaction, IHO1-axis association, DSB-factors recruitment on chromosome axes and DSB formation. The function of IHO1 C-terminus might be mediated by phosphorylation on S560 and S570. The scientific question of the manuscript is clear and the data are presented in a logical manner. However, some concerns shown below are suggested to be addressed.

Response: We are extremely happy that the reviewer has the opinion that our manuscript presents a well-designed study on a logical manner. We also greatly appreciate the constructive suggestions of the reviewer.

Remark

1. Line 119-120, the Y2H assay were conducted between different truncated IHO1 and HORMAD1. The interactions were not narrowed down to the HORMA domain. It is unappropriated to present “IHO1 fragments that included the C-terminal 75 amino acids of IHO1 interacted with HORMAD1 and, specifically, its HORMA domain”.

Response: We agree that the original statement was inaccurate. Whereas we identified the HORMA domain as the HORMAD1 region that interacted with the full length IHO1 we did not test if HORMA domain interacted with shorter versions of IHO1 (see Table S1 original manuscript). We performed further Y2H interactions for the revised manuscript. Results of the new Y2H experiments are included in a new Supplementary Table 1. We also added further details of the Y2H experiments to the table legend. These Y2H interactions support the original statement, hence we did not change the relevant section in the results.

Remark

2. In Fig 1B, what does "cross" mean? Does the cross indicate that the corresponding Y2H experiments were not conducted? If so, why the Y2H experiments corresponding to these "crosses" were not conducted? On the contrary, there are no “cross” shown in Fig. S2C. Please show the data in a consistent way.

Response: We thank the reviewer for pointing out these inconsistencies in the presentation of the data. We added crosses to Supplementary Fig. 3, which corresponds to Fig S2 in the previous version of the manuscript. The crosses show bait–prey combinations that were not tested in the Y2H assays as they were irrelevant for the scientific questions we addressed. For example, it was sufficient to perform Y2H assays between bait and prey versions of IHO1 that had the same C-terminal mutation to test if mutations in the C-terminus of IHO1 disrupted IHO1-IHO1 interactions. Thus, it was not necessary to test interactions between distinct versions of IHO1 that have distinct C-terminal mutations.

Remark

3. In Fig 1B, the panel and legend does not match.

Response: We are unclear what the reviewer is referring to. We speculate that Figure 1B was not sufficiently explained in the first version of the manuscript. Figure 1B shows yeast cultures after 3 or 2 days of growth on drop out plates. To improve comprehensibility, we expanded the description of this figure panel in the revised manuscript, and we indicate that “X” refers to bait-prey pairs that were not examined.

Remark

4. Line126-130, it is better to confirm the relationship between HORMAD1 and IHO1_WT, IHO1_S-T_AA or IHO1_A-A_SS using eukaryotic cell system, to verify the importance of S569 and S570 in mediating the interaction between HORMAD1 and IHO1.

Response: We used Y2H initially to allow rapid screening for regions and residues that are important for IHO1-HORMAD1 interactions. We agree that ectopic expression of various versions of IHO1 and HORMAD1 would have been a valid method to complement Y2H assays in testing the importance of S569 and S570 in IHO1-HORMAD1 interactions. We considered the suggested approach at the start of our analysis, however we opted against it because, like Y2H assays, eukaryotic protein expression systems are highly non-physiological as they do not reproduce the meiotic environment. Instead of ectopic protein expression in eukaryotic cell systems, we carried out overexpression of various versions of IHO1 in spermatocytes with the help of *in vivo* electroporation of testes (Fig 1C-D). This approach examined the behavior of IHO1 in their physiological meiotic environment and allowed us to test the localization of IHO1 versions, which is the most relevant assay for understanding how IHO1 C-terminus contributes to axial accumulation of DSB-promoting factors. Given the matching results of Y2H and *in vivo* electroporation experiments (highlighted in lines 166-168), in our opinion, expression of IHO1 versions in a mammalian culture system would be superfluous without providing new insights. While the proposed approach would provide little or no benefit it would require considerable effort and lengthy optimization (e.g.: previously, we observed toxicity of strong HORMAD1 overexpression in mouse somatic cell cultures, requiring optimization of transfection methods and identification of suitable cell lines/culture systems). Therefore, we kindly ask that our well-considered decision be accepted, based on the above arguments.

Remark

5. Line189, it presented that phosphorylation of the IHO1 C-terminus enhances axial accumulation. Axial accumulation of IHO1 relies on the interaction between IHO1 and

HORMAD1. Whether the interaction between IHO1 and HORMAD1 is affected by phosphorylation on S569 and/or S570 should be confirmed.

Response: We agree that the current data strongly suggests, but does not definitively prove, that phosphorylation on S569 and/or S570 enhances IHO1-HORMAD1 interaction. We invested considerable efforts (and also attempted the suggested experiment) to address the role of phosphorylation on S569 and/or S570. However, due to technical difficulties, we could not uncover the role and the mechanism of phosphorylation of S569 and/or S570 with absolute certainty (for details, please, see response to remarks 3-6 of Reviewer 2). Therefore, we toned down our conclusions and provide a more nuanced interpretation of the experiments that addresses the role of IHO1 phosphorylation (see changes in abstract (lines 36-38), and lines 168-174, 222-226, 243-246 and 560-582). We note that understanding the role of IHO1 phosphorylation is not necessary for the main message of the manuscript. The main message is that IHO1-HORMAD1 interaction and ANKRD31 represent complementing pathways that differentially promote seeding and growth of the DSB machinery, respectively.

Remark

6. Deletion of C-terminus of IHO1 affected the recruitment of MEI4 and REC114 to chromosome axis. Does the deletion of C-terminus of IHO1 affect the protein expression level of MEI4 and REC114? Can IHO1 C7Δ/C7Δ form complex with MEI4 and REC114 outside the chromosome axis?

Response: We found that MEI4 and REC114 levels are not affected significantly by the deletion of the last 7 amino acids of IHO1 as measured by western blot. This data is included in the revised manuscript in Supplementary Fig. 6A-B and lines 267-269.

Regarding IHO1 complex formation with MEI4 and/or REC114, our coIP experiments suggest that IHO1 efficiently forms complex with REC114 in conditions where IHO1 is not efficiently recruited to axis (*Iho1*^{C7Δ/C7Δ} and *Hormad1*^{-/-}, Fig. 1E, MEI4 was not tested in the coIPs experiments, because REC114, but not MEI4, interacts directly with IHO1). This conclusion is also consistent with results of *in vitro*/Y2H protein interaction assays. Our Y2H assays show that IHO1 interacts REC114 independent of axis if ectopically expressed in yeast (Fig. 1A-B). Finally, a recent publication showed complex formation of IHO1 N terminus (1-281 amino acids) with MEI4 and REC114 *in vitro*, suggesting that IHO1-REC114-MEI4 is able to form complexes in the absence of axis (Laroussi H,Kadlec J, 2023 EMBO J, doi: 10.15252/embj.2023113866. Epub 2023 Jul 11.).

Remark

7. Fig. 5G, Fig. S3A-B, Fig. 3E-F, Fig. 4I-J, all the assays were conducted in two independent experiments. In each assay, how many mice were analyzed?

Response: In each assay one mouse was included from each genotype. Thus, two independent experiments mean two mice of each genotype. We included this information in the figure legends of the revised manuscript. The mouse numbers are also indicated in the source file.

Remark

8. Line 212-213, Should “Fig 2” be “Fig 2D”?

Response: Thank you for pointing out this inaccuracy, we corrected the reference to the figure.

Remark

9. Some formats should be consistent throughout the manuscript. For instance, “2 fold” 、 “twofold” or “2-fold”?

Response: We corrected the formatting throughout the manuscript and made it consistent.

Remark

10. Line 885, 913, 1027, the text should be bold.

Response: Thank you for pointing out these formatting errors, we corrected the text as suggested.

Remark

11. The Results section is relatively verbose with lots of discussions. For instance, it might be better to move line 363-373 to the Discussion section.

Response: We agree that the description of the Results is complex, yet we think that it helps the reader if the possible interpretations and the direct conclusions of the experiments are presented in the Results section. This is the feedback we received from several colleagues who read the manuscript. Regarding the lines between 363-373 (lines 389-393 in the revised text), we think that this is a relatively minor point in the manuscript that does not justify a dedicated discussion paragraph. A dedicated paragraph in the discussion would distract from the main message of the manuscript, make a long discussion even longer and would disrupt the flow of the discussion. Therefore, we prefer to keep this paragraph in the Results.

** See Nature Portfolio’s author and referees’ website at www.nature.com/authors for information about policies, services and author benefits.

This email has been sent through the Springer Nature Tracking System NY-610A-NPG&MTS

Confidentiality Statement:

This e-mail is confidential and subject to copyright. Any unauthorised use or disclosure of its contents is prohibited. If you have received this email in error please notify our Manuscript Tracking System Helpdesk team at <http://platformsupport.nature.com> .

Details of the confidentiality and pre-publicity policy may be found here <http://www.nature.com/authors/policies/confidentiality.html>

Privacy Policy | Update Profile

DISCLAIMER: This e-mail is confidential and should not be used by anyone who is not the original intended recipient. If you have received this e-mail in error please inform the sender and delete it from your mailbox or any other storage mechanism. Springer Nature America, Inc. does not accept liability for any statements made which are clearly the sender's own and not expressly made on behalf of Springer Nature America, Inc. or one of their agents.

Please note that neither Springer Nature America, Inc. or any of its agents accept any responsibility for viruses that may be contained in this e-mail or its attachments and it is your responsibility to scan the e-mail and attachments (if any).

REVIEWERS' COMMENTS

Reviewer #1 (Remarks to the Author):

In the revised manuscript, the authors satisfied all the questions I raised. The manuscript should be open for the field.

Reviewer #2 (Remarks to the Author):

The authors have done a very good job at addressing the reviewer comments. I gladly recommend acceptance of the manuscript.

Reviewer #3 (Remarks to the Author):

The authors have addressed my concerns.